# Multibandgap quantum dot ensembles for solar-matched infrared energy harvesting

Bin Sun [1], Olivier Ouellette [1], F. Pelayo García de Arquer[1], Oleksandr Voznyy [1], Younghoon Kim[1,5], Mingyang Wei[1], Andrew H. Proppe [1,2,3], Makhsud I. Saidaminov[1], Jixian Xu[1], Mengxia Liu[1], Peicheng Li[4], James Z. Fan[1], Jea Woong Jo[1], Hairen Tan [1], Furui Tan[1], Sjoerd Hoogland[1], Zheng Hong Lu [4], Shana O. Kelley [2,3] & Edward H. Sargent [1]

As crystalline silicon solar cells approach in efficiency their theoretical limit, strategies are being developed to achieve efficient infrared energy harvesting to augment silicon using solar photons from beyond its 1100 nm absorption edge. Herein we report a strategy that uses multi-bandgap lead sulfide colloidal quantum dot (CQD) ensembles to maximize short-circuit current and open-circuit voltage simultaneously. We engineer the density of states to achieve simultaneously a large quasi-Fermi level splitting and a tailored optical response that matches the infrared solar spectrum. We shape the density of states by selectively introducing larger-bandgap CQDs within a smaller-bandgap CQD population, achieving a 40 meV increase in open-circuit voltage. The near-unity internal quantum efficiency in the optimized multi-bandgap CQD ensemble yielded a maximized photocurrent of $3.7 \pm 0.2\,\mathrm{mA\,cm^{-2}}$. This provides a record for silicon-filtered power conversion efficiency equal to one power point, a 25% (relative) improvement compared to the best previously-reported results.

[1] Department of Electrical and Computer Engineering, University of Toronto, 10 King's College Road, Toronto, ON M5S 3G4, Canada. [2] Department of Pharmaceutical Science, Leslie Dan Faculty of Pharmacy, University of Toronto, Toronto, ON M5S 3G4, Canada. [3] Department of Biochemistry, Faculty of Medicine, University of Toronto, Toronto, ON M5S 3M2, Canada. [4] Department of Material Science and Engineering, University of Toronto, 184 College St, Toronto, ON M5S 3E4, Canada. [5] Present address: Convergence Research Center for Solar Energy, Daegu Gyeongbuk Institute of Science and Technology, Daegu 42988, Republic of Korea. These authors contributed equally: Bin Sun, Olivier Ouellette, F. Pelayo García de Arquer. Correspondence and requests for materials should be addressed to E.H.S. (email: ted.sargent@utoronto.ca)

Photovoltaics accounted for 1.3% of the global energy supply in 2016, a number that is projected to increase to 20% by 2050[1]. As crystalline silicon (cSi) solar cells approach their theoretical efficiency limit[2], complementary strategies that further improve efficiency – without introducing significant additional cost – provide avenues to lower further the price of solar electricity.

With an indirect bandgap of 1.1 eV corresponding to an absorption edge at 1100 nm, Si solar cells leave up to 20% of the solar power reaching the Earth's surface unabsorbed. Efficient infrared energy harvesting that could complement Si absorption is a promising route to achieve broadband solar energy conversion, which is predicted to offer up to 6% additional power points on top of existing cSi photovoltaic solutions[3,4].

Colloidal quantum dots (CQDs) combine facile and broad spectral tunability via quantum-size tuning[5,6] with inexpensive manufacturing arising from their solution-processing. In the last decade, intensive efforts have focused on improving CQD synthesis, surface passivation, film formation, and device engineering; and these have led to great strides in increasing the performance of CQD photovoltaics[6–12]. IR CQD solar cells, on the other hand, have remained comparatively underexplored, and best IR-filtered PCEs lie below 0.5%[4,13,14].

An acute challenge in CQD solar cells is to realize simultaneously high short-circuit current ($J_{SC}$) and high open-circuit voltage ($V_{OC}$). As the size of QDs is increased and their bandgap shrinks so that more IR photons can be absorbed – a crucial step to harvest the solar power beyond 1100 nm − $V_{OC}$ decreases due to the smaller bandgap and the presence of energy losses ($E_{loss}$). $E_{loss}$ is defined as the deficit in $V_{OC}$ compared to the detailed balance limit for $V_{OC}$ at a given bandgap[15,16], and in CQD photovoltaics it stems primarily from bandtail states and recombination at defects. While energy losses on the order of 0.1–0.2 eV are observed for highly crystalline and low-defect materials such as cSi, CQDs are characterized by significantly higher values, reaching 0.4 eV[8,17,18]. The reduction of bandtail states to decrease this detrimental loss has therefore been a widespread theme in recent work[9]. The absorption/extraction compromise, which limits the thickness of the CQD active layer to a few hundreds of nanometers, represents an additional impediment to harvesting fully the infrared portion of the solar spectrum[19].

In this work, we revisit the conditions under which $V_{OC}$ is pinned in CQD ensembles. In doing so, we find a regime wherein $V_{OC}$ – rather than being rapidly pinned by the lowest bandgap component in a quantum dot ensemble[20] – is instead related linearly to the bandgap of the ensemble constituents. In this regime, the $V_{OC}$ for a given bandgap can be increased by the judicious addition of a larger bandgap species that modifies the density of states. We exploit this phenomenon and design CQD multi-bandgap ensembles that, by virtue of a tailored density of states and by spectrally matching the IR solar spectrum, simultaneously attain for the first time high $V_{OC}$ and high $J_{SC}$ of 0.4 V and 3.7 ± 0.2 mA cm$^{-2}$, respectively, more than 30% higher than previously reported values for both parameters[14]. As a result, we achieve cSi-filtered PCE of 1% – a record in infrared CQD PV[4,14].

## Results

**$V_{OC}$ modulation in multi-bandgap quantum dot ensembles.** Under illumination, the electron quasi-Fermi level increase in solar cells made from a single population of CQDs is dictated by the excited carrier density that can be sustained in the conduction band in steady state[8]. The overlap of the Fermi-Dirac occupation function at the quasi-Fermi level $f(E, E_{QFL})$ and the density of states (DOS) at the CQD conduction $g_{CB}(E)$ band determines this photoexcited electron density (Fig. 1a):

$$\Delta n = \int_{E_C}^{\infty} f(E, E_{QFL}) g_{CB}(E) \, \mathrm{d}E \tag{1}$$

A similar expression holds for photoexcited holes in the valence band. Mixing different CQD ensembles can be used to modify proportionally the effective DOS, which affects the overlap with the Femi-Dirac distribution of electrons depending on the relative weight of the populations and the difference in energy $\Delta E$ of the mixed dot ensembles (Fig. 1b). For a given photoexcited charge density $\Delta n$, $E_{QFL}$ will therefore increase if the relative density of lower energy states is reduced. We note that the F-D distribution is appropriate to describe the occupation

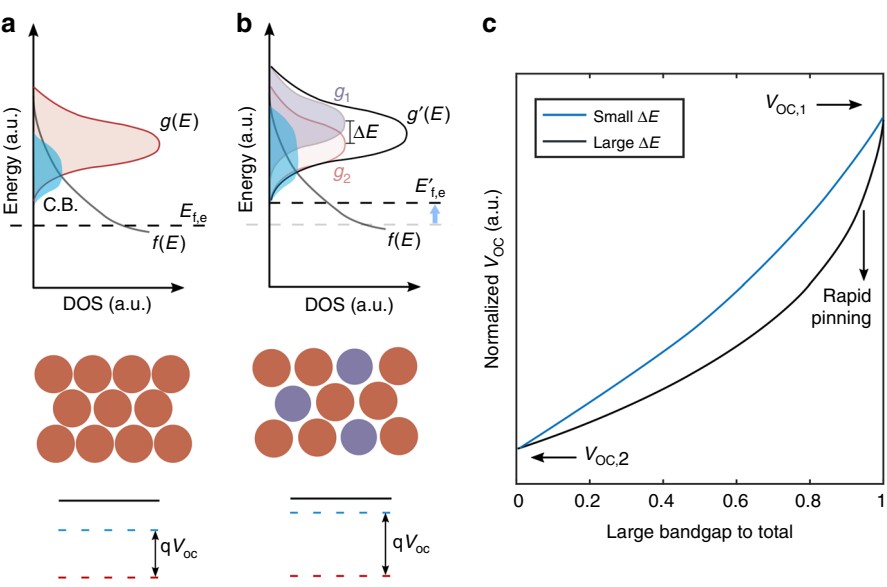

**Fig. 1** Open-circuit modulation in multi-bandgap QD ensembles under illumination. **a** A single population of small bandgap CQDs; **b** overlap of the Fermi-Dirac occupation function at the quasi-Fermi level $f(E,E_{QFL})$ and the density of states (DOS) at the CQD conduction $g_{CB}(E)$ band; **c** $V_{OC}$ behavior upon CQD mixing depending on the energy offset of the large bandgap inclusions in the mixed CQD films

probability not only in bands, but also of discrete energy states – such as, for example, in the monoatomic ideal gas[21], and, more broadly, in systems with single-particle energy levels[22]. Fermi-Dirac statistics apply only if the particles in the system can reach thermal equilibrium. Using ultrafast transient absorption spectroscopy (Supplementary Fig. 7 and 8), we verified experimentally that, in the pure and mixed CQD mixes, photoexcited electrons and holes[23] thermalize to the nearby available states in a few nanoseconds, well before they are lost to recombination, and thus do reach thermal equilibrium within their band.

To quantify this effect, we employed a band-filling model[24,25] and calculated the impact of CQD size mixing on $V_{OC}$. The conduction and valence DOS were built assuming Gaussian CQD size distributions and using the following size-to-bandgap relation:[26]

$$E_{G} = \frac{1}{0.0252x^2 + 0.283x},$$ (2)

where $E_{G}$ is the bandgap (eV) and $x$, the quantum dot diameter (nm). To retrieve the quasi-Fermi level splitting, which corresponds to the upper $V_{OC}$ limit, the steady state photoexcited charge generation rate is set equal to the recombination rate, which is assumed to be dominated by mid-gap tail states[27]. Details and calculation parameters are given in Supplementary Note 1 and Supplementary Fig. 1 and 2.

Different regimes are identified in the $V_{OC}$ behavior upon CQD mixing (Fig. 1c) as a function of the energy offset. When $\Delta E$ is large compared to the FWHM of the DOS (given by the size distribution), the open-circuit voltage is rapidly pinned to the $V_{OC}$ of the smallest-bandgap population. This case represents the conventional scenario in which, in a CQD film, the presence of narrow bandgap outliers and deep tail states dramatically reduces $V_{OC}$. As $\Delta E$ diminishes and the broadened DOS overlaps progressively more with $f(E)$, the open-circuit voltage shows an almost linear dependence on the $V_{OC}$ corresponding to the individual populations of the CQD ensemble. We therefore predict that modifying the DOS by mixing in CQDs with a slightly higher bandgap should have an appreciable beneficial effect on $V_{OC}$.

**Transport characteristics of multi-bandgap CQD ensembles.** We then proceeded to make films of CQD ensembles based on a solution-phase exchange method to replace the as-synthesized oleic acid capped CQDs with short inorganic halide ligands. Our solution exchange is based on a previously-reported protocol[28] for 1150 nm (large bandgap, L) and 1250 nm (small bandgap, S) CQDs. We optimized the solution exchange protocol as follows:[14,28] for 1150 nm CQDs, we kept PbI$_2$ and Pb(SCN)$_2$ at the same concentration as our previous work and modified the concentration of ammonium acetate (AA) from 10 mM to 60 mM in dimethylformamide (DMF) (Supplementary Fig. 3). When we increase the AA concentration, $V_{OC}$ decreases while FF and PCE increase before decreasing as well, which is ascribed to surface passivation and change in residual OA on the surface. The optimal concentration of AA of 20 mM was found for the 1150 nm CQD ligand exchange. We also optimized the 1250 nm CQD ligand exchange (Supplementary Fig. 4) by adjusting the AA concentration and added butylamine (BTA) to assist ligand exchange. In this case, the optimal concentration was found experimentally to be 60 mM for AA and 40 mM for BTA. We additionally performed X-ray photoelectron spectroscopy (XPS) to study the surface passivation (Supplementary Fig. 4c). The addition of BTA allows for more organics (oleic acid) and iodide ions to remain on the CQD surface, as indicated by the higher ratio of I:S and C:S compared to the control ligand exchange

without BTA[17,29]. We finally mixed the individual solutions (with the choice of ratio explored throughout this work) prior to CQD film formation.

To characterize the charge mobility and density of tail states for different quantum dot ensembles, we carried out field-effect transistor (FET) measurements (Fig. 2)[30]. We employed a bottom-gate top-contact configuration (Fig. 2a). The FET transfer characteristics for all the studied mixtures reveal the characteristic $n$-type character of halide-treated CQD films (Fig. 2b).

We retrieved the density of in-gap states from the measured transfer characteristics. By analyzing the exponential increase of the drain current below $V_{TH}$, which corresponds to transport through in-gap states, we obtain the density of in-gap states. The tail state distribution is calculated using the following equation:[28]

$$N_{td} = \left[ \left( \frac{S \cdot e}{kT \cdot \ln(10)} - 1 \right) \cdot \frac{C_i}{e} \right]^2 \cdot \epsilon_0 \epsilon_r^{-1}$$ (3)

where $S$ is the sub-threshold swing, the slope of the gate voltage vs. the log drain current between turn-on voltage and $V_{TH}$ that defines the boundary between the subthreshold and transport regime; $\epsilon_0$ is the vacuum permittivity; $\epsilon_r$ is the electric constant of the film, estimated to be 10.9[31]. After integrating the tail state distribution between the subthreshold and transport regime as shown in Fig. 2c for the mixture (weight ratio of 2 to 1), we obtain the density of tail states ($N_T$) (Supplementary Fig. 5) plotted in Fig. 2d. The pure large gap CQD film exhibits a $N_T$ of $1.5 \pm 0.2 \times 10^{16}$ cm$^{-3}$ (Supplementary Fig. 5), which is close to that of solution exchanged 950 nm PbS CQDs[28]. The pure small-gap CQD film shows a two orders of magnitude lower $N_T$ of $2.6 \pm 0.5 \times 10^{14}$ cm$^{-3}$ compared to the pure large gap CQD film (Fig. 2d), a finding we ascribe to better surface passivation. We also compared the transport properties of small bandgap dots exchanged with and without the BTA additive (Supplementary Fig. 6). The CQD film exchanged without BTA exhibits a $N_T$ of $5.2 \pm 0.4 \times 10^{16}$ cm$^{-3}$, while the addition of BTA lead to a much lower $N_T$ of $2.6 \pm 0.5 \times 10^{14}$ cm$^{-3}$, again due to better surface passivation. The CQD mixtures containing 33, 50, and 67% of large bandgap CQDs exhibit a $N_T$ of $2.8 \pm 0.4 \times 10^{15}$, $3.6 \pm 0.3 \times 10^{15}$, and $1.7 \pm 0.3 \times 10^{15}$ cm$^{-3}$, respectively, an order of magnitude lower than that of the pure large gap CQDs, indicating that the mixtures should have similar or even better carrier transport compared to the large bandgap CQD films.

In addition to obtaining tail density, we also extracted charge carrier mobility from FET measurements (Fig. 2d). The carrier mobility is calculated from the slope of $I_{DS}$ vs. $V_{GS}$ according to the equation $I_{DS} = \mu C_i \frac{W}{L} (V_{GS} - V_{TH}) V_{DS}$, where $\mu$ is the carrier mobility in the linear regime; $I_{DS}$ is the drain current; $L$ and $W$ are the channel length (50 μm) and channel width (2.5 mm) respectively; and $V_{GS}$ and $V_{TH}$ are the gate voltage and threshold voltage, respectively. The pure large-gap CQD film has an electron mobility of $0.052 \pm 0.003$ cm$^2$ V$^{-1}$ s$^{-1}$, while the pure small-gap CQD film shows a lower mobility of $0.020 \pm 0.002$ cm$^2$ V$^{-1}$ s$^{-1}$, which may be due to the residual oleic acid ligands on the CQD surface. The CQD films with inclusions of large bandgap CQDs of 33, 50, and 67% exhibit mobilities of $0.026 \pm 0.004$, $0.023 \pm 0.004$, and $0.021 \pm 0.003$ cm$^2$ V$^{-1}$ s$^{-1}$, respectively. In addition, we studied charge carrier transport between the two differently-sized distributions using ultrafast transient absorption spectroscopy (Supplementary Fig. 7 and 8). We found that the wide size dispersity allows for photoexcited charges to be thermally excited into larger and/or smaller dots, thereby thermalizing into the nearby available states in a few nanoseconds. We also conducted ultraviolet photoelectron spectroscopy (UPS) (Supplementary Fig. 9) to determine the position of the

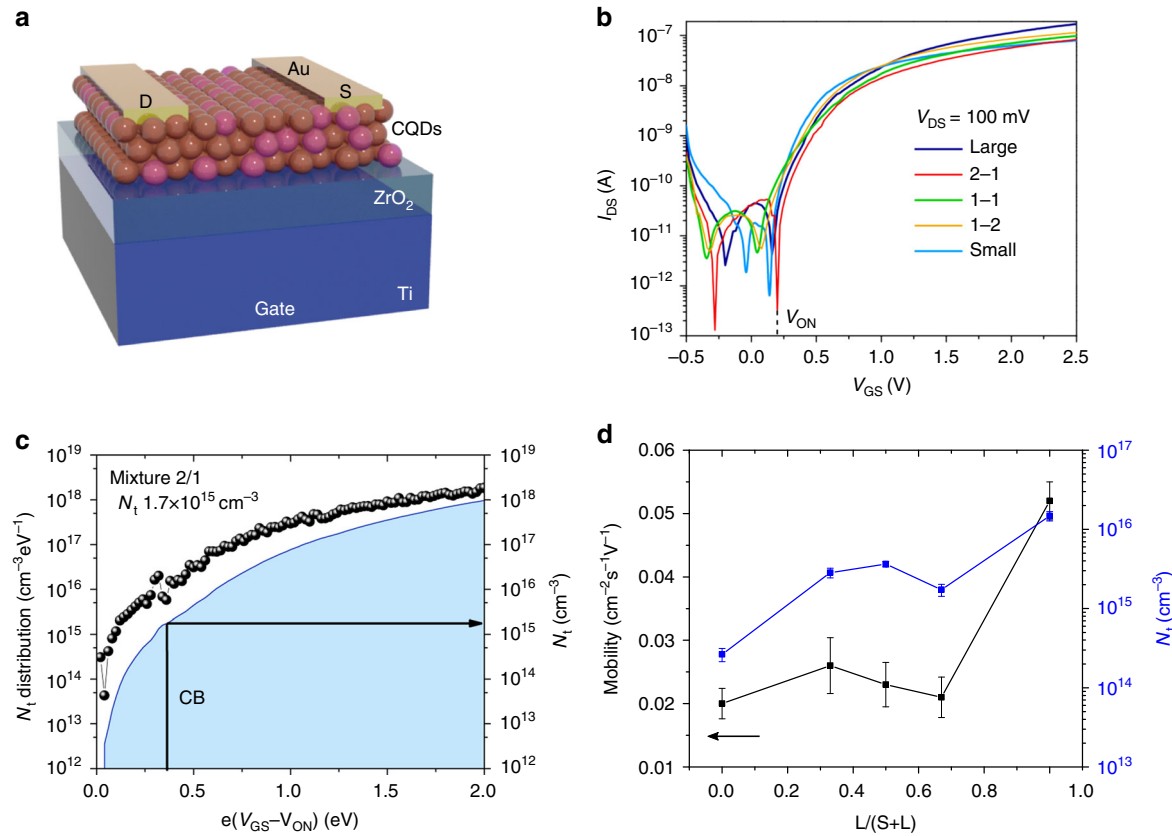

**Fig. 2** Transport properties of CQD multi-bandgap ensembles **a** bottom-gate top-contact field-effect transistor structure; transfer characteristics of pure CQD and multi-bandgap CQD ensembles with different weight ratio of large bandgap (L) to small bandgap (S) CQDs showing onset voltage ($V_{ON}$) **b**, transfer characteristics of pure and mixed CQDs with different weight ratios of large bandgap (L) to small bandgap (S); **c** tail state density ($N_T$) of the optimal CQD mixture (weight ratio of 2 to 1) as a function of gate bias as calculated with Eq. (3); **d** mobility and trap density as a function of the inclusion of L in the mixed films

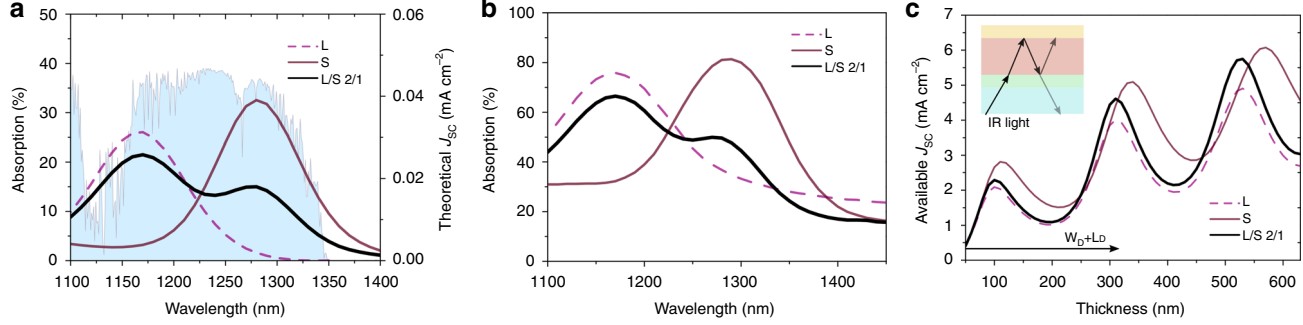

**Fig. 3** Expected $J_{SC}$ from multi-bandgap CQD ensembles. absorptance measured from (**a**) the CQD films on glass and **b** complete solar cell devices including the gold electrode mirror; **c** calculated $J_{SC}$ as a function of CQD film thickness

energy levels of the single size CQDs, and confirmed that they have energy levels needed for band alignment.

**Tailoring the multi-bandgap CQD ensembles spectral response**. The band-filling model and FET analysis indicate that the mixtures can achieve improved $V_{OC}$ and comparable charge transport properties. We sought to leverage this property and turned our attention to the optical behavior of the multi-bandgap CQD ensemble and aimed to maximize the overlap of light absorption with the cSi-filtered infrared solar spectrum.

Figure 3a shows the single pass absorptance of CQD films of the same thickness (300 ± 10 nm) on optical glasses, where the 2:1

(large bandgap: small gap) films have a lower absorptance maxima than pure CQDs (~30%). The mixtures do not show significantly higher IR photon absorption than the pure CQD films. In a complete CQD solar cell, however, the gold back-electrode serves also as a mirror. The resulting reflection contributes to the device absorptance and introduces resonant absorption. This is due to interference between the forward-propagating light from the illuminated side and the backward-propagating light reflected on the gold electrode and can be controlled and optimized by adjusting the active layer thickness[13,32]. We thus measured the total absorption through complete PV devices (Fig. 3b). We observed that light absorption

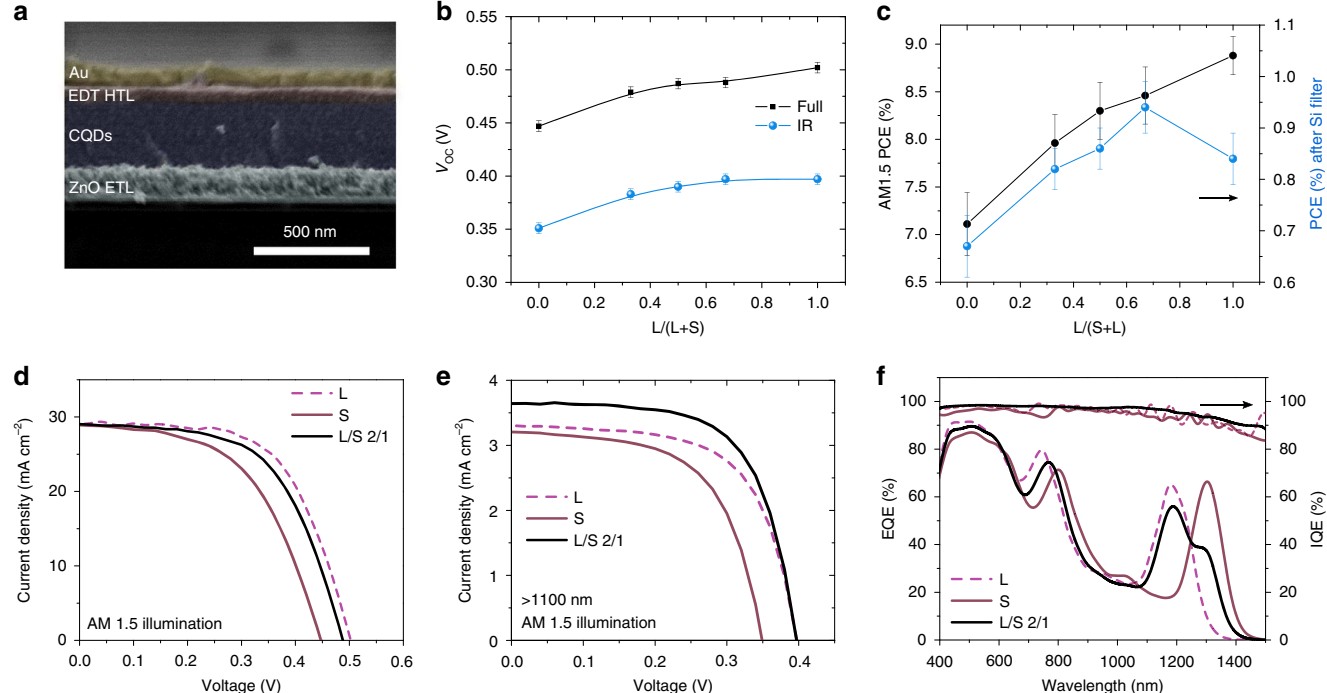

**Fig. 4** PV Device architecture and performance. **a** Device architecture and cross-sectional SEM image of the best mixed CQD film solar cell. Measured (**b**) $V_{OC}$ and **c** PCE with different inclusions of large bandgap CQDs. **d** J-V characteristics under AM1.5 G, **e** J-V characteristics after 1100 nm; **f** EQE curves and IQE curves of optimal single and mixed CQD solar cell devices

in the mixtures is enhanced at certain wavelengths, which contribute to additional photo-generated current. To confirm the effect of optical resonance, we additionally measured light absorption in CQD films before Au deposition (Supplementary Fig. 10), which lack the resonant absorption peaks present in the absorption spectra of devices containing the Au back mirror, thus confirming the role of the resonant mechanism.

To optimize the total IR absorption, we calculated the available $J_{SC}$ as the thickness of the active layer varies using the transfer-matrix method[32,33] (Fig. 3c and Supplementary Fig. 11). Pure large bandgap dots and 2 to 1 mixture films have a local $J_{SC}$ maximum at a thickness of about 300 nm, while the pure small bandgap dots can absorb more light at about 340 nm, which is due to the absorption peak position difference. The available $J_{SC}$ decreases after the first local maximum, and much thicker CQD films (above 500 nm) are required for a net increase in $J_{SC}$. For such a large thickness, the efficiency of charge carrier extraction will be dramatically reduced, as the diffusion length in these CQD solids is in the order of hundreds of nm. Based on these findings, we narrowed our attention to 2 to 1 mixtures and active layer thicknesses ranging from 200 to 350 nm.

**PV device performance**. We then characterized the photovoltaic performance of solar cells employing multi-bandgap CQD ensembles (Fig. 4). The devices consist of a ZnO layer, acting as an electron acceptor; an active layer formed of PbS CQD ensemble; EDT-exchanged PbS CQDs as the hole acceptor, and thermally evaporated gold as the top electrode.

The open-circuit voltage shows the predicted trend upon quantum dot mixing (Fig. 4b). The AM1.5 $V_{OC}$ for large bandgap is 0.50 V, and 0.45 V for small bandgap CQDs. The $V_{OC}$ of 0.45 V for small-gap CQDs is higher than previous reports for similar sizes (0.38 V), which we ascribe to the lower $N_T$ stemming from better passivation. The $V_{OC}$ of mixtures gradually shifts between the two pure CQDs, relating to the weight inclusions almost linearly as expected from the state-filling model. We calculated

the energy loss dependence on the inclusion of large bandgap CQDs in mixed CQD films under AM1.5 irradiation (Supplementary Fig. 12) and found that the mixed CQDs exhibit the lowest $E_{loss}$ (<0.27 eV), lower than that of the large and small bandgap CQDs (0.33 and 0.30 eV, respectively).

We then characterized the PV devices after an 1100 nm long-pass filter to replicate the effect of a silicon front cell. The mixture with 67% of large bandgap CQDs shows an IR $V_{OC}$ of 0.40 V, similar to that of pure large bandgap CQDs films. This further demonstrates the benefit of multi-bandgap CQD ensembles to maximize open-circuit voltage. With fewer inclusions of large-gap CQDs, the IR $V_{OC}$ of the mixtures gradually decreases with the decreased portion of large-gap CQDs. The similar IR $V_{OC}$ of mixed CQD films compared to pure large bandgap CQD films can be attributed to the lower $N_T$ than that of pure large bandgap CQD films, which reduces trap-assisted recombination, lowering the drop of $V_{OC}$ with the reduced light intensity. We also investigated the impact of a higher bandgap difference between the mixed CQDs on the resulting $V_{OC}$ (Supplementary Fig. 13). The $V_{OC}$ of mixes of CQDs with exciton peaks at 1150 nm and 1512 nm is quickly pinned to that of the small bandgap CQDs, in agreement with the theoretical model.

Multibandgap CQD ensembles exhibit a superior IR PCE compared to pure CQD films (Fig. 4c, Supplementary Table 3). The best IR PCE of 0.95 ± 0.04% was obtained in the mixture containing 67% large bandgap CQDs, with a 0.40 ± 0.01 V $V_{OC}$, 3.7 ± 0.2 mA cm$^{-2}$ $J_{SC}$, and a 65 ± 1% fill factor (FF). The best large-bandgap CQD films, on the other hand, led to a PCE of 0.84 ± 0.03% with $V_{OC}$, $J_{SC}$, and FF at 0.40 ± 0.01 V, 3.3 ± 0.2 mA cm$^{-2}$, 64 ± 1%; the small bandgap CQD solar cells yielded a PCE of 0.67 ± 0.05% with $V_{OC}$, $J_{SC}$, and FF at 0.35 V, 3.2 ± 0.2 mA cm$^{-2}$, 60 ± 1%. The device performance under unfiltered AM1.5 G illumination is presented in Supplementary Fig. 18 and Supplementary Table 2 for reference.

We tested three different multi-bandgap CQD ensemble configurations, containing large bandgap CQDs from 33 to

67%; all these three compositions showed at least 20% improvement compared to the small bandgap samples. The enhancement of absorption in mixtures containing 67% large-bandgap CQDs yields an enhanced $J_{SC}$ of 3.7 mA cm$^{-2}$, calculated from the EQE:

$$J_{SC} = q \int_0^\infty \mathrm{EQE}(\lambda)\gamma_i(\lambda)\mathrm{d}\lambda$$

where $\gamma_i(\lambda)$ is the incident solar photon flux spectrum. Tailoring the absorption spectrum leads to this increase in $J_{SC}$ by better matching the external quantum efficiency (EQE) spectrum to the solar spectrum over the 1100 nm to 1400 nm spectral range (Supplementary Fig. 14). The EQE of the best mixed CQD device is wider than its pure counterparts, as seen by the increase in full-width half-maximum (FWHM) of the exciton peak (Supplementary Fig. 15), which in turn leads to an increase in photocurrent when the absorption spectrum is well matched to the solar spectrum. The shape of the exciton peak and its FWHM was tuned to the solar spectrum to increase $J_{SC}$ while minimizing $V_{OC}$ loss. We note that the extended FWHM of the exciton peak did not improve $J_{SC}$ under full-AM1.5-spectrum one-sun conditions (Fig. 4d and Supplementary Fig. 18) because optical resonances improve in some spectral regions, but decrease in others, the absorbance.

We also calculated the internal quantum efficiency IQE using the measured EQE and simulated light absorption in the CQD active layer (Fig. 4f). Multibandgap CQD ensembles show enhanced EQE and IQE compared to pure CQD films, as transport of photogenerated charges takes place mainly through low defect-density, small-bandgap CQD paths. The enhanced EQE in multi-bandgap CQD ensembles shows not only the improved spectral range from the extended absorption, but also the enhanced transport, higher than pure CQD films, as was demonstrated by FET results.

We finally investigated the thickness-dependent performance of the pure and mixed CQD films (Supplementary Fig. 16) The optimal thickness for every device is found to be around 300 nm, where $J_{SC}$ decreases as the thickness increases due to resonant absorption as discussed above, which is in good agreement with the double pass absorption and simulation in Fig. 3 and Supplementary Fig. 10. For different inclusions of L, the 67% of L CQDs yields the highest $J_{SC}$ of 3.7 mA cm$^{-2}$ at a thickness of 300 nm, whereas the 50 and 33% of L CQDs both yield the highest $J_{SC}$ of 3.4 cm$^{-2}$ when they are 320 nm thick.

## Discussion

In this work, we report a strategy based on multi-bandgap CQD ensembles to achieve high open-circuit voltage, short-circuit current and PCE in cSi-filtered IR photovoltaics. We engineered the DOSs in this platform to improve quasi-Fermi level splitting and increase $V_{OC}$. We further leveraged the optical properties of multi-bandgap CQD ensembles to achieve solar-matched IR light absorption, leading to high $J_{SC}$ and a record cSi-filtered power conversion efficiency of 1%, setting a record for silicon-filtered CQD PVs. This strategy, which allows decoupling of the traditional $V_{OC} - J_{SC}$ trade-off, has the potential to raise the IR PCE in the direction of the 6% theoretical limit with the improved light absorption properties of a mixture of CQD populations well-matched to the solar spectrum.

## Methods

**Materials and characterization**. The oleate-capped PbS CQDs and ZnO nanoparticles were synthesized following our previous reports[24]. Other chemicals were obtained from commercial suppliers and used as is. Optical absorption measurements were performed on a Lambda 950 UV-Vis-IR spectrometer.

**QDs ligand exchange and solution preparation**. The PbI$_2$/Pb(SCN)$_2$/AA DMF solution ligand exchange is carried out in a test tube in air. Precursor solution (PbI$_2$ 0.1 M, AA 0.02 M for 1150 nm CQDs, and PbI$_2$ 0.1 M, BTA 0.04 M, and AA 0.06 M for 1250 nm CQDs) is dissolved in DMF. 0.5 ml of oleate-capped PbS CQDs octane solution (50 mg ml$^{-1}$) was added to 5 ml of precursor solution, followed by vigorously mixing for 2 min until the CQDs completely transferred to the DMF phase. The DMF phase was then washed three times with octane. Then 1150 nm CQDs precipitated during the exchange, while 1250 nm CQDs are stable in DMF and precipitated by adding 4 mL of acetone. The CQD precipitates were collected by centrifugation, followed by vacuum drying for 15 min. The CQDs were redispersed in a mixture of BTA and DMF at a volume ratio of 8/2 (250 mg ml$^{-1}$) for film by spin coating.

**FET fabrication**. Bottom-gate top-contact FET configuration is used as follows: 70 nm of titanium gate was thermally evaporated onto a glass substrate, followed by 15 nm of ZrO$_2$ as a dielectric layer using atomic layer deposition (ALD). After 300 °C baking for 1 h, the pre-exchanged QDs dissolved in BTA/DMF were spin-coated onto the substrate. Then 70 nm of Au source/drain electrodes were thermally deposited using an Angstrom Engineering Amod deposition system. Agilent 4155c semiconductor analyzer was used to characterize the FET devices.

**CQD solar cell fabrication**. ZnO layer was adopted as electron acceptor layer and formed on ITO-coated glass substrate by spin coating the ZnO nanoparticles solution at 3000 rpm for 30 s. Then PbS CQDs (pure CQDs or mixtures with different weight ratio), 250 mg mL$^{-1}$ in BTA/DMF (8/2 volume ratio) solution, were spin cast on ZnO substrate at 2500 RPM for 30 s, followed by two layers of EDT-exchanged PbS CQDs as follows: 2 drops of oleic acid-capped PbS CQDs octane solution (50 mg mL$^{-1}$) were spin coated at 2500 rpm for 10 s, followed by soaking in 0.01% EDT in acetonitrile (ACN) solution for 30 s and washing with ACN for 3 times. For the top electrode, 120 nm of Au was deposited on EDT PbS CQD film to complete the device.

**External and internal quantum efficiency**. EQE and IQE spectra were acquired on a QuantX-300 quantum efficiency measurement system (Newport). Mono-chromated white light from a xenon lamp was mechanically chopped at a frequency of 25 Hz. EQE spectra were acquired at zero electrical bias, whereas IQE spectra were calculated from an EQE spectra taken at a negative bias of −2 V using the following formula: IQE = EQE(0 V)/EQE(−2V)[34].

**Current-voltage under simulated AM1.5**. The current-voltage behavior under a simulated AM1.5 solar spectrum was acquired and corrected according to EQE spectra. Devices were kept in an inert N$_2$ atmosphere. The input power density was adjusted to 1 Sun using a NIST-traceable calibrated reference cell (Newport 91150 V). To account for the spectral mismatch between the AM1.5 G reference spectrum and the spectrum of the lamp, a current density correction factor was used for each device, corresponding to the ratio of the value calculated from integrating the EQE spectrum and the value measured under illumination[35]. The lamp spectrum was measured using irradiance-calibrated spectrometers (USB2000 and NIR512, Ocean Optics) and is shown in Supplementary Fig. 17. The calculated spectral mismatch factors are shown in Supplementary Table 3.

**Ultrafast transient absorption spectroscopy**. A regeneratively amplified Yb:KGW laser (PHAROS, Light Conversion) laser was used to generate femtosecond pulses (250 fs FWHM) at 1030 nm as the fundamental beam with a 5 kHz repetition rate. This fundamental beam was passed through a beam-splitter, where one arm was used to pump an optical parametric amplifier (ORPHEUS, Light Conversion) for the narrowband pump, and the other arm was focused into a sapphire crystal (Ultrafast Systems) in order to generate a NIR white-light continuum probe with a spectral window of 1050 nm to 1600 nm. Both arms were directed into a commercial transient absorption spectrometer (Helios, Ultrafast Systems). The probe pulse was delayed relative to the pump pulse to provide a time window of up to 8 ns. All measurements were performed using an average power of 100 μW with a spot size of 0.40 μm$^2$, assuming a Gaussian beam profile.

## Data availability

All relevant data are available from the authors upon request.

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

## Acknowledgements

This work was supported by Ontario Research Fund-Research Excellence program (ORF7-Ministry of Research and Innovation, Ontario Research Fund-Research Excellence Round 7), and by the Natural Sciences and Engineering Research Council (NSERC) of Canada. M.I.S. acknowledges the support of the Banting Postdoctoral Fellowship Program, administered by the Government of Canada.

## Author contributions

B.S., O.O., F.P.G.A., S.H., S.O.K. and E.H.S. designed and directed this study. B.S., Y.H.K., M.W., A.H.P., M.I.S., J.X., M.L., P.L., J.Z.F., J.W.J., H.T., F.T. and Z.H.L. carried out the synthesis, ligand exchange, characterizations, and solar-cell device fabrication. O.O., F.P. G.A. and O.V. performed the simulation studies. B.S., O.O., F.P.G.A., O.V. and E.H.S. wrote the manuscript.

## Additional information

**Competing interests:** The authors declare no competing interests.

