## [Peer Review File · Nature Communications]

Reviewers' comments:

Reviewer #1 (Remarks to the Author):

The authors report the conditions under which Voc is pinned in CQDs ensembles, and reported that Voc is related linearly to the bandgap of the ensemble constituents. The manuscript provides a useful guide line for enhancing Voc in solar cells using CQDs without sacrificing Jsc. However, there are several things to be addressed by the authors.

#1.

L46-54: Energy loss or Voc loss is an important figure of merit to evaluate solar cell performance, and is a core topic for the manuscript. In the introduction part, the authors mention that solar cells using highly crystalline and low-defect materials give energy losses on the order of 0.1–0.2 eV (L46-53). These energy losses seem much smaller than estimated by theoretical analysis. In theory, the smallest energy loss for the single junction solar cells working under the 1-sun condition at room temperature is approximately 0.4 eV or so [e.g. J. Appl. Phys. 51, 4494 (1980)].

The most highly efficient c-Si solar cell [Prog. Photovolt: Res. Appl., 23, 1 (2015)] gives a Voc of 0.740 V, yielding an energy loss of 0.36 V (1.1 eV - 0.74 eV), which is much larger than the values cited in the authors' manuscript. The energy loss of the best performing single junction GaAs solar cell is also in the same range.

The authors must include the corresponding reference to validate the energy loss values (0.1 to 0.2 eV) in the manuscript.

#2. Figure 4(e):

In Figure 4 (e), the EQE curves in the shorter wavelengths (< 1100 nm) should be plotted. The EQE curves plotted in the figure appears to be measured without a 1100 nm long-pass filter because the solar cells show 20-30% EQE values even at 1100 nm.

If this is the case, the EQE curves of the solar cells with the filter should be included in the figure. The transmittance and/or absorptance spectra of the 1100nm long-pass filter that the authors used in the experiments must be also included in the manuscript or supporting information.

#3. L240 - 246:

The authors achieved the best IR PCE in the mixture containing 67% large bandgap CQDs.

The manuscript must include the PCE, Voc, Jsc, and FF together with the energy loss for the three types of CQD solar cells (L, S, and L+S) without the 1100 nm long-pass filter.

#4. L275 - 278:

"The strategy, which allows to decouple the traditional Voc - Jsc tradeoff, has the potential to raise the IR PCE towards the 6% theoretical limit as the charge transport properties in CQD solids are further improved."

The authors were successful in decoupling the traditional Voc - Jsc trade-off by introducing so-called resonant absorption mechanism. The effectiveness of this mechanism depends heavily on CQD layer thickness. Towards the 6% theoretical limit, not only charge transport properties but also light absorption efficiency have to be improved even further. In doing so, the thickness of CQD layer can't be arbitrarily chosen so as to use the resonant absorption mechanism in an effective way. Is the resonant mechanism available to develop such a high performance solar cell?

Reviewer #2 (Remarks to the Author):

This manuscript reports the multibandgap quantum dot ensembles for solar cells using infrared energy harvesting (for the wavelength region of larger than 1100 nm). The authors mixed two kinds of PbS QDs with different sizes in the CQD active layer and discussed the dependence of the photovoltaic performances on the weight of the two kinds QDs. The best VOC and best JSC are 0.4 V and 3.8 mA cm⁻² for the IR region, respectively. The authors used a model based on the overlap of the Fermi-Dirac occupation function of the two kinds of QDs to calculate the quasi-Fermi levels of electrons and holes to discuss the results. However, there is a fundamental problem in the theoretical model. So I cannot recommend it to be published in Nature Communications at the present situation. The following are the detailed comments.

(1) The Fermi-Dirac occupation function is for bulk with band structures and not suitable for QDs because the energy levels are separated with each other due to the quantum confinement effects of the QDs. Especially, in the case of PbS, the sizes of the QDs (3-5 nm) is very smaller compared to the Bohr diameter (36 nm), so the quantum confinement effects are very strong. The authors used an overlay of the Fermi-Dirac occupation functions of the two different size QDs to calculate DOS or Nt. However, the two different size QDs are mixed randomly in the layer. So why the authors can separate these two kinds QD films with each other? The consideration is not correct here.

(2) The dependence of the mobility on the inclusion of L in the mixed films (Figure 2(d)) should be discussed quantitatively.

(3) The defect density of different mixed QD films should be quantitatively shown and discussed. What is the proof of the value of the trap density used in the calculation shown in Table S1?

(4) What is the working principle of the solar cell with the mixed QDs active layer? How can carrier transport through the QDs with different sizes?

(5) Figure 4(c), Jsc dependence on the L/(L+S) should be discussed quantitatively.

(6) The authors said that "We selectively modified the solution exchange precursor by adding butylamine to individually control the exchange rate, leaving more OA organic ligand on smaller bandgap CQD surface seeking to minimize their tail states. We then mixed the individual solutions (with the choice of ratio explored throughout this work) prior to CQD film formation." However, I cannot find the details in the experimental part. The details of the experiments and the quantitative OA ligand on the different bandgap CQDs should be shown.

Reviewer #3 (Remarks to the Author):

The present work proposes a mixing band gap strategy for enhancing the quantum dot IR solar cell performance. The paper is about an interesting and important issue but is somewhat lack of novelty and premature. The idea of introducing different size of QDs to improve the IR PV performance is good. However, more experimental data and improved consistency to explain the fundamental principles under this improvement are required. Therefore, the paper is not suitable to be published in Nature Communications. When the authors resubmit or submit to another journal, they may need to consider the following comments/questions.

1. Mixing different bandgap QDs to enhance the IV performance is not genuine. There are many previous works have been published, and the authors are encouraged to comparing their novelty and scientific breakthrough with previous works such as Adv. Optical Mater. 2014, 2, 549–554; Adv. Funct. Mater. 2013, 23, 4149; Sci. Rep. 5, 2015, 10626.

2. The authors claim a new record in infrared QDPV. Is there any certificate can be provided to support this achievement? And it is required that more details about the QDPV evaluation set-up such as input power density and spectrum details after IR filter.
3. The bandgap difference between the two QDs selected in this paper is 1.08eV and 1.00eV. Therefore, the energy difference is only 0.08eV, and the wavelength difference is only 100nm. Did the author perform any other size QD comparison which has a slightly bigger difference in optical bandgap? If there is no other combination can be found superior to this combination, it may indicate the strategy of mixing different size QD for improving the IR PV performance is not promising due to the space for further improvement is extremely small (i.e. tweaking band gap difference within 0.08eV).
4. Moreover, if the optical band gap difference is crucial, the authors are encouraged to provide the parameters such as FWHM extracted from single size QDs and mixed QDs. A rationalization relationship between FWHM difference and the device IR PCE enhancement shall also be provided in the manuscript.
5. In Figure 1b, how can the author determine the photogeneration rates between these two size QDs are equal? In Figure 1a and b, the diagram is constructed under dark or illumination?
6. Did the author perform any UPS measurement to determine the energy level difference between single size QDs and mixed size QDs? A measured energy level difference shall consistent with the simulation results.
7. The mobility values in Figure 2d are not consistent with the description in line 169 and line 172. In Fig2 d, is there rational explanation or supporting data why small QD has lower tail state density? And Authors claim that the charge transport behavior remains unaffected due to the continuous network of small QDs. However, mobility data in Figure2d looks fluctuated and even, proportional to tail state density (subthreshold width data). Authors need to clarify this inconsistency.
8. Did the author perform any dark diode analysis on these devices with a different combination of the QDs?
9. The current size determination is only calculated from an empirical model which may overestimate the size difference under such a small band gap difference. Further systematic microscopy analysis may support the simulation work, particular in the case of the ensemble.
10. In Figure 3a and Figure 3b, the absorption of small band gap QDs are higher than mixed QD system, particular in the onset region. However, in Figure 4f, the IQE onset of mixed QD system is much higher than small bandgap QDs. Furthermore, in Figure 4e and 4f, why the large band gap QDs exhibit much higher IQE than the others in the wavelength region of 1200nm to 1300nm? The authors are encouraged to explain these controversial phenomena.
11. The author mentioned that "the resulting reflection contributes to the device absorptance and introduces resonant absorption mechanisms" Any further experimental evidence or theoretical explanation for this claim? The author should provide more evidence and explanation of the difference absorption enhancement phenomena between L, S and L/S (2: 1) layer with and without the Au electrode because it seems to be directly related to enhanced performance of mixed QDPV.
12. Based on the calculation in Figure 3c, the optimum thickness of each S, L and S+L film is different. Therefore, it might be more reasonable to compare the QDPV performance with each optimum condition of the QD films. Related to this comment, there is no thickness dependent Jsc data in figure S1a (line 263-264).
13. There are some typo errors;
In the abstract, the first sentence seems to be 'As crystalline silicon (cSi) solar cells approach in their theoretical efficiency limit,' not 'As crystalline silicon (cSi) solar cells approach in efficiency their theoretical limit,'
missing punctuations (Line 79, 86, etc.),
the redundant abbreviations (line 75).
In line 197, "Figure 2c" should be "Figure 3c"

Reviewer #4 (Remarks to the Author):

The manuscript by Sargent and coworkers reports about QD solar cells with band gap lower than the silicon band gap displaying efficiency (when filtered by the silicon band gap) around 1%. The authors claim that by mixing QDs of different sizes, they can limit the damage on the Voc, and they can still get some relatively good transport, as in their case the larger particles have better quality than the smaller one. To be frank all this will be totally superfluous if all the QDs size will be of equal quality, as the authors seems to acknowledge as well.

More technically, as there are several authors who reported in the past solar cells made with the particles of the size the authors are investigating, which are not enormously large, will be very interesting to see the characteristics of the devices measured with the full AM1.5 solar illumination and not only the filtered devices.

In the figure 4c, I do not understand why the larger QDs exhibit the lower performances. The authors will answer that they have more traps, as the want to prove with the transistor works, but what I find peculiar is the discontinuity in the trend. I would like to suggest them to test also some different size of QDs, maybe what they have found is a singularity in the QD size behavior. Overall, I do not think this manuscript in the current status, is publishable here or somewhere else, as it rises many doubts on what the authors are really observing.

Reviewer #1 (Remarks to the Author):

The authors report the conditions under which V_{oc} is pinned in CQDs ensembles, and reported that V_{oc} is related linearly to the bandgap of the ensemble constituents. The manuscript provides a useful guide line for enhancing V_{oc} in solar cells using CQDs without sacrificing J_{sc} . However, there are several things to be addressed by the authors.

Thank you for the valuable suggestions. We have acted on each concern as documented in the point-by-point following:

#1.

L46-54: Energy loss or V_{oc} loss is an important figure of merit to evaluate solar cell performance, and is a core topic for the manuscript. In the introduction part, the authors mention that solar cells using highly crystalline and low-defect materials give energy losses on the order of 0.1–0.2 eV (L46-53). These energy losses seem much smaller than estimated by theoretical analysis. In theory, the smallest energy loss for the single junction solar cells working under the 1-sun condition at room temperature is approximately 0.4 eV or so [e.g. *J. Appl. Phys.* 51, 4494 (1980)].

The most highly efficient c-Si solar cell [*Prog. Photovolt: Res. Appl.*, 23, 1 (2015)] gives a V_{oc} of 0.740 V, yielding an energy loss of 0.36 V (1.1 eV - 0.74 eV), which is much larger than the values cited in the authors' manuscript. The energy loss of the best performing single junction GaAs solar cell is also in the same range.

The authors must include the corresponding reference to validate the energy loss values (0.1 to 0.2 eV) in the manuscript.

We have corrected the definition of energy loss in the manuscript to:

“[...] the deficit in V_{OC} compared to the detailed balance limit for V_{OC} at a given bandgap (*J. Appl. Phys.* 32, 510–519 (1961), *Sol. Energy* 130, 139–147 (2016))”.

#2. Figure 4(e):

In Figure 4 (e), the EQE curves in the shorter wavelengths (< 1100 nm) should be plotted. The EQE curves plotted in the figure appears to be measured without a 1100 nm long-pass filter because the solar cells show 20-30% EQE values even at 1100 nm.

If this is the case, the EQE curves of the solar cells with the filter should be included in the figure. The transmittance and/or absorbance spectra of the 1100nm long-pass filter that the authors used in the experiments must be also included in the manuscript or supporting information.

In light of the referee's feedback, we reproduced devices presented in the manuscript and collected EQE spectra with and without the 1100 nm long-pass filter. Figure 4 has been updated accordingly. The transmittance spectrum of the 1100 nm long-pass filter is now included in Figure S14 of the SI.

#3. L240 - 246:

The authors achieved the best IR PCE in the mixture containing 67% large bandgap CQDs.

The manuscript must include the PCE, V_{oc} , J_{sc} , and FF together with the energy loss for the three types of CQD solar cells (L, S, and L+S) without the 1100 nm long-pass filter.

Figure S12 now presents the calculated energy loss for all device types. Their photovoltaic characteristics under non-filtered AM1.5G illumination have been added to Table S3. We added to the main text:

“We calculated the energy loss dependence on the inclusion of large bandgap CQDs in mixed CQD films under AM1.5 irradiation (Figure S12) and found that the mixed CQDs exhibit the lowest E_{loss} (less than 0.27 eV), lower than that of the large and small bandgap CQDs (0.33 and 0.30 eV, respectively).”

#4. L275 - 278:

“The strategy, which allows to decouple the traditional V_{oc} - J_{sc} tradeoff, has the potential to raise the IR PCE towards the 6% theoretical limit as the charge transport properties in CQD solids are further improved.”

The authors were successful in decoupling the traditional V_{oc} - J_{sc} trade-off by introducing so-called resonant absorption mechanism. The effectiveness of this mechanism depends heavily on CQD layer thickness. Towards the 6% theoretical limit, not only charge transport properties but also light absorption efficiency have to be improved even further. In doing so, the thickness of CQD layer can't be arbitrarily chosen so as to use the resonant absorption mechanism in an effective way. Is the resonant mechanism available to develop such a high performance solar cell?

In light of the referee's feedback, we have added an extended version of Figure 3c in the Supplementary Information, which shows the calculated J_{SC} for active layer thicknesses reaching 3 μm . As the thickness is increased, a greater proportion of the incident light is absorbed on the first pass, reducing the role of the resonance. However, the resonance allows J_{SC} to reach its maximum value at comparatively lower active layer thicknesses, well before light absorption saturates to the value limited by parasitic absorption. Optical resonance is therefore an important mechanism that must be taken into consideration in IR-CQD devices on their way to high efficiency.

Figure S11: Available J_{SC} in thick active layers, illustrating the role of optical resonance in enhancing light absorption.

Reviewer #2 (Remarks to the Author):

This manuscript reports the multibandgap quantum dot ensembles for solar cells using infrared energy harvesting (for the wavelength region of larger than 1100 nm). The authors mixed two kinds of PbS QDs with different sizes in the CQD active layer and discussed the dependence of the photovoltaic performances on the weight of the two kinds QDs. The best VOC and best JSC are 0.4 V and 3.8 mA cm⁻² for the IR region, respectively. The authors used a model based on the overlap of the Fermi-Dirac occupation function of the two kinds of QDs to calculate the quasi-Fermi levels of electrons and holes to discuss the results. However, there is a fundamental problem in the theoretical model. So I cannot recommend it to be published in Nature Communications at the present situation. The following are the detailed comments.

(1) The Fermi-Dirac occupation function is for bulk with band structures and not suitable for QDs because the energy levels are separated with each other due to the quantum confinement effects of the QDs. Especially, in the case of PbS, the sizes of the QDs (3-5 nm) is very smaller compared to the Bohr diameter (36 nm), so the quantum confinement effects are very strong. The authors used an overlay of the Fermi-Dirac occupation functions of the two different size QDs to calculate DOS or N_t . However, the two different size QDs are mixed randomly in the layer. So why the authors can separate these two kinds QD films with each other? The consideration is not correct here.

We now better explain in the revised manuscript that the F-D distribution is appropriate to describe the occupation probability not only in bands, but also of discrete energy states – such as, for example, in the monoatomic ideal gas [Rend. Lincei 3, 145-149 (1926)].

We also now more clearly point out that derivations of the F-D distribution often begin with discrete energy levels, such as those of a quantum dot. For instance, the derivation found in “Solid-State Physics” by Ibach and Lüth begins by considering “an atomic system with single-particle energy levels”, a broad definition that applies well to QDs.

We also now better explain that a condition to allow the use of F-D statistics is that the particles in the system must be in thermal equilibrium. In the revised manuscript, we use ultrafast transient absorption spectroscopy (Figure S7 and S8) and find that charge carriers in the CQD films do reach thermal equilibrium within their band before being lost to recombination. Figure S7a and S7b show the 2D spectra (bottom left), temporal cross-sections (top left) and spectral cross-sections (right) for pure small (S) and large (L) bandgap dot ensembles, respectively. Negligible carrier population can be seen away from the photoexcited species, as evidenced in the temporal cross-sections. In the case of the 2:1 mix, however, significant charge transfer is observed from L to S following photoexcitation of L (Fig. S8a) and from S to L following photoexcitation of S (Fig. S8b). This indicates that photoexcited charge carriers thermalize to the nearby available states in a few nanoseconds, well before they recombine. In terms of charge carrier occupation distribution, the mixed CQD films therefore behave like a semiconductor whose density of states near the band-edge corresponds to a sum of both CQD species' DOS. We also now explain in the Supplementary Information that we build the modeled density of states of the mixes by using a weighed sum of their Gaussian distributions. This is justified by the fact that the QD sizes are indeed mixed randomly in the film, allowing for the resulting DOS to be modeled by a combination of both

sizes' respective DOS.

Figure S7. Transient absorption spectra of pure CQD films. Bottom left: 2D spectrum. Top left: Temporal cross-section. Right: Spectral cross-section. (a) Pure small bandgap CQD film, photoexcitation at 1300 nm. (b) Pure large bandgap CQD film, photoexcitation at 1160 nm.

Figure S8. Transient absorption spectra of L/S 2/1 mixed CQD films. Bottom left: 2D spectrum. Top left: Temporal cross-section. Right: Spectral cross-section. (a) Photoexcitation in the small bandgap population at 1300 nm. (b) Photoexcitation in the large bandgap population at 1160 nm. Charge transfer from the photoexcited population to the other is observed in both cases.

We have added the following to the Methods, under “Ultrafast Transient Absorption Spectroscopy:”

“A regeneratively amplified Yb:KGW laser (PHAROS, Light Conversion) laser was used to generate femtosecond pulses (~ 250 fs FWHM) at 1030 nm as the fundamental beam with a 5 kHz repetition rate. This fundamental beam was passed through a beam-splitter, where one arm was used to pump an optical

parametric amplifier (ORPHEUS, Light Conversion) for the narrowband pump, and the other arm was focused into a sapphire crystal (Ultrafast Systems) in order to generate a NIR white-light continuum probe with a spectral window of ~1050 - 1600 nm. Both arms were directed into a commercial transient absorption spectrometer (Helios, Ultrafast Systems). The probe pulse was delayed relative to the pump pulse to provide a time window of up to 8 ns. All measurements were performed using an average power of 100 μW with a spot size of 0.40 μm^2 , assuming a Gaussian beam profile.”

- (2) The dependence of the mobility on the inclusion of L in the mixed films (Figure 2(d)) should be discussed quantitatively.

In light of the referee’s feedback, we added the following discussion:

“The pure large-gap CQD film has an electron mobility of $0.052\pm 0.003 \text{ cm}^2\text{V}^{-1}\text{s}^{-1}$, while the pure small-gap CQD film shows a lower mobility of $0.020\pm 0.002 \text{ cm}^2\text{V}^{-1}\text{s}^{-1}$, which may be due to the residual oleic acid ligands on the CQD surface. The CQD films with inclusions of large bandgap CQDs of 33%, 50%, and 67% exhibit mobilities of 0.026 ± 0.004 , 0.023 ± 0.004 , and $0.021\pm 0.003 \text{ cm}^2\text{V}^{-1}\text{s}^{-1}$, respectively. The mixtures have similar mobilities with the films of pure small-gap CQDs, suggesting that charge carriers find a path through small-bandgap dots..”

- (3) The defect density of different mixed QD films should be quantitatively shown and discussed. What is the proof of the value of the trap density used in the calculation shown in Table S1?

We have added the quantitative FET analysis to Figure 2 and Figure S5 and the following discussion on trap density:

“We retrieved the density of in-gap states from the measured transfer characteristics. By analyzing the exponential increase of the drain current below V_{TH} , which corresponds to transport through in-gap states, we obtain the density of in-gap states. The tail state distribution is calculated using the following equation²⁵:

$$N_{td} = \left[\left(\frac{S \cdot e}{kT \cdot \ln(10)} - 1 \right) \cdot \frac{C_i}{e} \right]^2 \cdot \epsilon_0 \epsilon_r^{-1}$$

where S is the sub-threshold swing, the slope of the gate voltage versus the log drain current between turn-on voltage and V_{TH} that defines the boundary between the subthreshold and transport regime; ϵ_0 is the vacuum permittivity; ϵ_r is the electric constant of the film, estimated to be 10.9²⁷. After integrating the tail state distribution between the subthreshold and transport regime as shown in Figure 2c for the mixture (weight ratio of 2 to 1), we obtain the density of tail states (N_T) (Figure S5) plotted in Figure 3d. The pure large gap CQD film exhibits a N_T of $1.5\pm 0.2 \times 10^{16} \text{ cm}^{-3}$ (Figure S5), which is close to that of solution exchanged 950 nm PbS CQDs²⁵. The pure small-gap CQD film shows a two orders of magnitude lower N_T of $2.6\pm 0.5 \times 10^{14} \text{ cm}^{-3}$ compared to the pure large gap CQD film (Figure 2d), a finding we ascribe to better surface passivation. We also compared the transport properties of small bandgap dots exchanged with and without the BTA

additive (Figure S6). The CQD film exchanged without BTA exhibits a N_T of $5.2 \pm 0.4 \times 10^{16} \text{ cm}^{-3}$, while the addition of BTA lead to a much lower N_T of $2.6 \pm 0.5 \times 10^{14} \text{ cm}^{-3}$, again due to better surface passivation. The CQD mixtures containing 33%, 50%, and 67% of large bandgap CQDs exhibit a N_T of $2.8 \pm 0.4 \times 10^{15}$, $3.7 \pm 0.3 \times 10^{15}$, and $1.7 \pm 0.3 \times 10^{15} \text{ cm}^{-3}$, respectively, an order of magnitude lower than that of the pure large gap CQDs, indicating that the mixtures should have similar or even better carrier transport compared to the large bandgap CQD films.”

In addition, we have updated the calculation to increase transparency. The value used in the calculation of Figure 1c, shown in Table S1, has been updated to 10^{16} cm^{-3} , a conservative value extracted from the FET analysis. Figures S2a and S2c have also been added to illustrate that trap density used does not affect the resulting trend in V_{OC} pinning, but rather only impact the V_{OC} limit.

Figure S1. Effect of trap density on the V_{OC} model. (a) Calculation done with two different trap densities, illustrating how the V_{OC} pinning trend from Figure 1c is not affected. (b) V_{OC} limit for different trap densities in absolute unit, showing that only the magnitude of V_{OC} is changed.

(4) What is the working principle of the solar cell with the mixed QDs active layer? How can carrier transport through the QDs with different sizes?

Charge transport is understood to take place through low-bandgap-CQDs percolation channels within the films, as evidenced by the fact that the charge carrier mobility of the mixed CQD films is close to that of the pure low bandgap CQD films (see data in point 2).

In addition, we have performed ultrafast transient absorption spectroscopy on the quantum dot films to investigate the energy distribution of charge transfer between the dot populations and have added the results to Figures S7 and S8. This experiment illustrates that photoexcited charge carriers have access to the neighbouring CQDs despite possible size (and thus energy) differences between them, with a more favourable pathway towards lower energy states.

(5) Figure 4(c), J_{sc} dependence on the $L/(L+S)$ should be discussed quantitatively.

In light of the referee's feedback, we made new devices to confirm the J_{sc} dependence of the $L/(S+L)$ ratio. We reproduced the best devices in addition to devices with different thickness, and now report their J_{sc} in Figure S17, with the following discussion:

“The CQD film with 67% L inclusions gives the highest J_{sc} of 3.7 mA cm^{-2} at an active layer thickness of 300 nm , whereas the 50% and 33% mixes yield the highest J_{sc} of 3.4 and 3.4 mA cm^{-2} at a thickness of 320 nm . These IR J_{sc} are higher than the best J_{sc} from the single CQD films, which can be attributed in part to the wider FWHM (Figure S15).”

(6) The authors said that “We selectively modified the solution exchange precursor by adding butylamine to individually control the exchange rate, leaving more OA organic ligand on smaller bandgap CQD surface seeking to minimize their tail states. We then mixed the individual solutions (with the choice of ratio explored throughout this work) prior to CQD film formation.” However, I cannot find the details in the experimental part. The details of the experiments and the quantitative OA ligand on the different bandgap CQDs should be shown. We now provide a full discussion in the main text on the solution exchange modification process and have added Figure S3 and S4 to complement it:

“We optimized the solution exchange protocol as follows:^{12,25} for 1150 nm CQDs, we kept PbI_2 and $\text{Pb}(\text{SCN})_2$ at the same concentration as our previous work and modified the concentration of ammonium acetate (AA) from 10 mM to 60 mM in dimethylformamide (DMF) (see Figure S3). When we increase the AA concentration, V_{oc} decreases while FF and PCE increase before decreasing as well, which is ascribed to surface passivation and change in residual OA on the surface. The optimal concentration of AA of 20 mM was found for the 1150 nm CQD ligand exchange. We also optimized the 1250 nm CQD ligand exchange (Figure S4) by adjusting the AA concentration and added butylamine (BTA) to assist ligand exchange. In this case, the optimal concentration was found experimentally to be 60 mM for AA and 40 mM for BTA. We additionally performed X-ray photoelectron spectroscopy (XPS) to study the surface passivation (Figure 3c). The addition of BTA allows for more organics (oleic acid) and iodide ions to remain on the CQD surface, as indicated by the higher ratio of I:S and C:S compared to the control ligand exchange without BTA^{15,26}.”

We also performed the FET measurement on BTA-assisted ligand-exchanged films shown in Figure S6, where the CQD film without BTA shows an electron mobility of $0.0044 \text{ cm}^2\text{s}^{-1}\text{V}^{-1}$, which is one order of magnitude lower than the one with BTA. The lower mobility is attributed to surface traps, which is also calculated to be $5.2 \pm 0.4 \times 10^{16} \text{ cm}^{-3}$, while the BTA-assisted film has a much lower surface trap density of $2.6 \pm 0.5 \times 10^{14} \text{ cm}^{-3}$, which is in good agreement with the PV device performance shown in Figure S4.

Reviewer #3 (Remarks to the Author):

The present work proposes a mixing band gap strategy for enhancing the quantum dot IR solar cell performance. The paper is about an interesting and important issue but is somewhat lack of novelty and premature. The idea of introducing different size of QDs to improve the IR PV performance is good. However, more experimental data and improved consistency to explain the fundamental principles under this improvement are required. Therefore, the paper is not suitable to be published in Nature Communications. When the authors resubmit or submit to another journal, they may need to consider the following comments/questions.

1. Mixing different bandgap QDs to enhance the IV performance is not genuine. There are many previous works have been published, and the authors are encouraged to comparing their novelty and scientific breakthrough with previous works such as Adv. Optical Mater. 2014, 2, 549–554; Adv. Funct. Mater. 2013, 23, 4149; Sci. Rep. 5, 2015, 10626.

We now cite the reference, Sci. Rep. 5, 2015, 10626, as ref. 18. We have shown in our work that when we mix different-bandgap QDs with a large bandgap difference (>0.1 eV), the V_{OC} is be rapidly pinned to that of the smallest bandgap component in the mixture (Figure S13), which is what is observed in the work of ref 18. In our work, we select the two CQD components with a small bandgap difference, which gives a relatively linear V_{OC} change according to the ensemble constituents. Furthermore, the mixing strategy provides better transport, which is in agreement with the ref 18. In our work, the improvement in J_{SC} at no cost to V_{OC} lead to an increase in IR PCE of $\sim 11\%$.

2. The authors claim a new record in infrared QDPV. Is there any certificate can be provided to support this achievement? And it is required that more details about the QDPV evaluation set-up such as input power density and spectrum details after IR filter.

We have now added to the Supplementary Information the solar simulator lamp spectrum (Figure S16) and the calculated spectral mismatch factors for each reported device (Table S2). We have also updated the “Current-Voltage under Simulated AM1.5” section of the Methods to provide more details on the spectral characterization and input power density adjustment.

We have not obtained a third-party certification. The previous best published results to which we compared our devices are 0.8% and 0.5% from references 4 and 12, which we have now added to the main text.

3. The bandgap difference between the two QDs selected in this paper is 1.08eV and 1.00eV. Therefore, the energy difference is only 0.08eV, and the wavelength difference is only 100nm. Did the author perform any other size QD comparison which has a slightly bigger difference in optical bandgap? If there is no other combination can be found superior to this combination, it may indicate the strategy of mixing different size QD for improving the IR PV performance is not promising due to the space for further improvement is extremely small (i.e. tweaking band gap difference within 0.08eV).

In light of the referee’s feedback, we have now fabricated devices with CQD ensembles having a larger bandgap difference: 1150 nm and 1500 nm (0.25 eV difference). The results have been added to Figure S13 and exhibit the expected V_{OC} pinning by the 1500 nm dots, in good agreement with the calculation in Figure 1.

Prior works have suffered a trade-off when working towards increasing both J_{SC} and V_{OC} . For IR PV, the AM1.5G spectrum has spectral gaps with low irradiation due to atmospheric absorption and single-size CQD ensembles with a sharp absorption peak cannot efficiently utilize the IR irradiation. Our strategy to selectively mix different size CQDs is promising to maximize J_{SC} by better matching the CQD absorption spectrum and the solar IR spectrum, without V_{OC} suffering from being pinned by the smallest bandgap component.

4. Moreover, if the optical band gap difference is crucial, the authors are encouraged to provide the parameters such as FWHM extracted from single size QDs and mixed QDs. A rationalization relationship between FWHM difference and the device IR PCE enhancement shall also be provided in the manuscript.

We have extracted the FWHM from the measured absorption spectra of CQD films and added the data to Figure S15. The mixed QDs show much larger FWHM compared to the single size QDs, which is directly related to the observed J_{SC} and PCE increase.

5. In Figure 1b, how can the author determine the photogeneration rates between these two size QDs are equal? In Figure 1a and b, the diagram is constructed under dark or illumination?

- We have now updated the calculation and Figure 1c to include the expected variation of the generation rate. The generation rate for both sizes of dots was calculated following the procedure described in the SI, and a weighted average was used as an estimate to model the mixes. The absorption coefficients used to calculate the generation rates are now shown in Figure S1. Finally, details were added in the SI to clarify the calculation method.

- We have now updated the figure caption to specify that Figures 1a,b represent a device under illumination.

6. Did the author perform any UPS measurement to determine the energy level difference between single size QDs and mixed size QDs? A measured energy level difference shall consistent with the simulation results.

In light of the referee's feedback, we carried out UPS measurements and provide the energy levels and UPS spectra in Figure S9, which indicated the VB difference is small, and the EF difference as well. The small difference ($< 0.1\text{eV}$) favourable for charge transport between the different CQD ensembles and consistent with the transient absorption spectra shown in Figure S8.

7. The mobility values in Figure 2d are not consistent with the description in line 169 and line 172. In Fig2 d, is there rational explanation or supporting data why small QD has lower tail state density? And Authors claim that the charge transport behavior remains unaffected due to the continuous network of small QDs. However, mobility data in Figure2d looks fluctuated and even, proportional to tail state density (subthreshold width data). Authors need to clarify this inconsistency.

- We have corrected the erroneous axis label.

- The lower tail state density has been obtained from FET measurements. We compared the small bandgap CQD with different ligand exchange conditions in Figure S6, and our optimized method yields higher mobility and lower tail state density, which we ascribe to

better surface passivation (confirmed by XPS in Figure S4). The same phenomena have been reported in *Adv Mater* 2017, 29, 1703627.

- We now discuss the mobility and tail state density quantitatively.

“We retrieved the density of in-gap states from the measured transfer characteristics. By analyzing the exponential increase of the drain current below V_{TH} , which corresponds to transport through in-gap states, we obtain the density of in-gap states. The tail state distribution is calculated using the following equation²⁵:

$$N_{td} = \left[\left(\frac{S \cdot e}{kT \cdot \ln(10)} - 1 \right) \cdot \frac{C_i}{e} \right]^2 \cdot \epsilon_0 \epsilon_r^{-1}$$

where S is the sub-threshold swing, the slope of the gate voltage versus the log drain current between turn-on voltage and V_{TH} that defines the boundary between the subthreshold and transport regime; ϵ_0 is the vacuum permittivity; ϵ_r is the electric constant of the film, estimated to be 10.9^{27} . After integrating the tail state distribution between the subthreshold and transport regime as shown in Figure 2c for the mixture (weight ratio of 2 to 1), we obtain the density of tail states (N_T) (Figure S5) plotted in Figure 3d. The pure large gap CQD film exhibits a N_T of $1.5 \pm 0.2 \times 10^{16} \text{ cm}^{-3}$ (Figure S5), which is close to that of solution exchanged 950 nm PbS CQDs²⁵. The pure small-gap CQD film shows a two orders of magnitude lower N_T of $2.6 \pm 0.5 \times 10^{14} \text{ cm}^{-3}$ compared to the pure large gap CQD film (Figure 2d), a finding we ascribe to better surface passivation. We also compared the transport properties of small bandgap dots exchanged with and without the BTA additive (Figure S6). The CQD film exchanged without BTA exhibits a N_T of $5.2 \pm 0.4 \times 10^{16} \text{ cm}^{-3}$, while the addition of BTA lead to a much lower N_T of $2.6 \pm 0.5 \times 10^{14} \text{ cm}^{-3}$, again due to better surface passivation. The CQD mixtures containing 33%, 50%, and 67% of large bandgap CQDs exhibit a N_T of $2.8 \pm 0.4 \times 10^{15}$, $3.7 \pm 0.3 \times 10^{15}$, and $1.7 \pm 0.3 \times 10^{15} \text{ cm}^{-3}$, respectively, an order of magnitude lower than that of the pure large gap CQDs, indicating that the mixtures should have similar or even better carrier transport compared to the large bandgap CQD films. [...]

The pure large-gap CQD film has an electron mobility of $0.052 \pm 0.003 \text{ cm}^2 \text{V}^{-1} \text{s}^{-1}$, while the pure small-gap CQD film shows a lower mobility of $0.020 \pm 0.002 \text{ cm}^2 \text{V}^{-1} \text{s}^{-1}$, which may be due to the residual oleic acid ligands on the CQD surface. The CQD films with inclusions of large bandgap CQDs of 33%, 50%, and 67% exhibit mobilities of 0.026 ± 0.004 , 0.023 ± 0.004 , and $0.021 \pm 0.003 \text{ cm}^2 \text{V}^{-1} \text{s}^{-1}$, respectively. The mixtures have similar mobilities with the films of pure small-gap CQDs, suggesting that charge carriers find a path through small-bandgap dots.”

8. Did the author perform any dark diode analysis on these devices with a different combination of the QDs?

We now report in the Supplementary Information that we measured the devices dark IVs and extracted the instantaneous ideality factor, as shown in the figure below. In the quasi-flat region, the ideality factor of small bandgap CQDs and mixed films is slightly lower than that of the large bandgap CQD films. This is an indication of a higher density of tail states in large bandgap CQDs, in good agreement with the FET data. The ideality factor increasing above 2 at higher voltages is due to series resistance.

Dark diode analysis of best PV device with different inclusion of large bandgap CQDs. | a) dark IV and b) instantaneous ideality factor extracted from dark IV.

9. The current size determination is only calculated from an empirical model which may overestimate the size difference under such a small band gap difference. Further systematic microscopy analysis may support the simulation work, particular in the case of the ensemble.

We now better explain that – to provide experimental support for the size distributions of the CQD ensembles and thus to provide a basis for the modeled DOS – we have instead relied on the absorption spectra of the pure and mixed CQDs. The Gaussian shape of the exciton peaks, their central positions, and FWHMs, all match those used in the theoretical model. In the case of the mixes, the presence of two distinct peaks effectively confirms the two underlying size distributions and their respective proportions. Furthermore, experiments have shown that simply using CQD absorption spectra can indeed yield the same information as extensive TEM measurements and image analysis [ACS Nano, 2012, 6(10), 9021-9032].

10. In Figure 3a and Figure 3b, the absorption of small band gap QDs are higher than mixed QD system, particular in the onset region. However, in Figure 4f, the IQE onset of mixed QD system is much higher than small bandgap QDs. Furthermore, in Figure 4e and 4f, why the large band gap QDs exhibit much higher IQE than the others in the wavelength region of 1200nm to 1300nm? The authors are encouraged to explain these controversial phenomena.

We have updated figure 4f with a new, more accurate measurement of IQE, now described as follows in the Methods:

“EQE and IQE spectra were acquired on a QuantX-300 quantum efficiency measurement system (Newport). Monochromated white light from a xenon lamp was mechanically chopped at a frequency of 25 Hz. EQE spectra were acquired at zero electrical bias, whereas IQE spectra were calculated from an EQE spectra taken at a negative bias of -2 V using the following formula: $IQE = EQE(0V) / EQE(-2V)$.”

The previous method relied on optical modelling to calculate parasitic absorption from other components of the solar cells, which was then subtracted from the measured absorption spectrum to obtain an estimate of the isolated absorption spectrum of the active layer, which was then used to calculate the IQE spectra shown in Figure 4f. The method we now used, made possible by the recent purchase of new equipment, is based on the extraction of all photoexcited charge carriers with a strong negative bias, and therefore allows for a more

accurate estimation of the charge collection efficiency. The reported IQE spectra now show a similar trend for all devices, and the discussion in the main text has been updated accordingly.

11. The author mentioned that “the resulting reflection contributes to the device absorptance and introduces resonant absorption mechanisms” Any further experimental evidence or theoretical explanation for this claim? The author should provide more evidence and explanation of the difference absorption enhancement phenomena between L, S and L/S (2:1) layer with and without the Au electrode because it seems to be directly related to enhanced performance of mixed QDPV.

We have now added details to the theoretical explanation of the resonant absorption mechanism in the main text:

“The resulting reflection contributes to the device absorptance and introduces resonant absorption. This phenomenon is due to interference between the forward-propagating light from the illuminated side and the backward-propagating light reflected on the gold electrode, and can be controlled and optimized by adjusting the active layer thickness [...] To confirm the effect of optical resonance, we additionally measured the absorption spectra CQD films before Au deposition (Figure S10), which lack the resonant absorption peaks present in the absorption spectra of devices containing the Au back mirror, thereby confirming the role of the resonant mechanism.”

The absorption spectra of CQD films with and without the Au back mirror have been added to the Supplementary Information (Figure S10). Resonant absorption peaks with a central wavelength which varies with the thickness of the CQD layer appear in the spectra after Au deposition, thereby confirming the presence of optical interference. The mixed CQD film benefits more from this mechanism than its pure counterparts because the shape of its exciton peak is better matched to the solar spectrum in the 1100 nm – 1350 nm spectral range.

12. Based on the calculation in Figure 3c, the optimum thickness of each S, L and S+L film is different. Therefore, it might be more reasonable to compare the QDPV performance with each optimum condition of the QD films. Related to this comment, there is no thickness dependent J_{sc} data in figure S1a (line 263-264).

We now provide the thickness dependent J_{sc} and PCE in figure S17 and added the following discussion to the main text:

“We finally investigated the thickness-dependent performance of the pure and mixed CQD films (Figure S1) The optimal thickness for every device is found to be around 300 nm, where J_{sc} decreases as the thickness increases due to resonant absorption as discussed above, which is in good agreement with the double pass absorption and simulation in Figure 3 and S10. For different inclusions of L, the 67% of L CQDs yields the highest J_{sc} of 3.7 mA cm⁻² at a thickness of 300 nm, whereas the 50% and 33% of L CQDs both yield the highest J_{sc} of 3.4 cm⁻² when they are 320 nm thick.”

13. There are some typo errors; In the abstract, the first sentence seems to be ‘As crystalline silicon (cSi) solar cells approach in their theoretical efficiency limit,’ not ‘As crystalline silicon (cSi) solar cells approach in efficiency their theoretical limit,’ missing punctuations

(Line 79, 86, etc.), the redundant abbreviations (line 75). In line 197, “Figure 2c” should be “Figure 3c”.

We have corrected these typos in the manuscript.

Reviewer #4 (Remarks to the Author):

The manuscript by Sargent and coworkers reports about QD solar cells with band gap lower than the silicon band gap displaying efficiency (when filtered by the silicon band gap) around 1%. The authors claim that by mixing QDs of different sizes, they can limit the damage on the Voc, and they can still get some relatively good transport, as in their case the larger particles have better quality than the smaller one. To be frank all this will be totally superfluous if all the QDs size will be of equal quality, as the authors seems to acknowledge as well.

We now provide the ligand-exchange optimization process details for small bandgap dots in S3-4, which was done with the aim of using equal-quality QDs to maximize performance. The mixing strategy in fact does not rely on the CQDs being of different quality, as is evidenced by the fact that the mixes exhibit higher performance than either individual constituent. Mixing CQDs for Si-filtered PV is a strategy that should always be beneficial for it allows to better match the CQD absorption spectrum with the specific shape of the Sun's spectrum beyond 1100 nm and increase photocurrent at no cost to V_{OC} .

More technically, as there are several authors who reported in the past solar cells made with the particles of the size the authors are investigating, which are not enormously large, will be very interesting to see the characteristics of the devices measured with the full AM1.5 solar illumination and not only the filtered devices.

We now provide the performance under unfiltered AM1.5 in Figure 4, Table S2 and Figure S18.

In the figure 4c, I do not understand why the larger QDs exhibit the lower performances. The authors will answer that they have more traps, as the want to prove with the transistor works, but what I find peculiar is the discontinuity in the trend. I would like to suggest them to test also some different size of QDs, maybe what they have found is a singularity in the QD size behavior. Overall, I do not think this manuscript in the current status, is publishable here or somewhere else, as it rises many doubts on what the authors are really observing.

In light of the referee's feedback, we have improved the quality of the larger CQD devices and fabricated more devices with new batches using 1150 nm and 1260 nm CQDs. By further optimizing the thickness, we obtained better large QD devices that show performance under IR illumination only slightly lower than the small QD devices. The mixed CQD devices indeed improve upon both of those devices, justifying the benefit of our mixing strategy. We have updated Figure 3 & 4 with the new data.

In addition, the field effect carrier mobility of our 950 nm (Adv. Mater., 2017, 29, 1700749), 1150 nm, and 1250 nm PbS CQD films are 0.07, 0.05, 0.02 $\text{cm}^2 \text{V}^{-1}\text{s}^{-1}$; the mobility decreases with increasing size, which is the opposite of what was observed in other work with thiol passivation (*Nano Letters*, 2010, 10, 1960). This should be due to the different CQD surface passivation.

We believe that the new experiments and analysis support these conclusions:

- Mixing CQD ensembles of different sizes allows us better to match the IR portion of the solar spectrum in a way that improves V_{OC} at no cost to J_{SC} . This was experimentally confirmed and further, has led to a new absolute record in Si-filtered CQD photovoltaics.

- To explain the fact that V_{OC} was not pinned to the smallest-bandgap component in the mix, as would be expected, we have built a theoretical model based on state-filling and trap-limited recombination. We have studied the cases of small and large bandgap differences and have now confirmed the conclusions of the model by preparing devices with a large bandgap difference of 0.26 eV (Figure S13), whose performance features the predicted V_{OC} pinning.

- To confirm that our new performance record was specifically due to the mixing strategy, we have first optimized the ligand exchange to obtain the highest-quality CQDs for both sizes used, with the details of the process now shown in Figure S3. In addition, we have factored-in the role and effect of optical resonance in the devices *via* transfer matrix modelling (Figure 3c, now extended in Figure S11) and have made sure that all devices were fabricated at their respective optimal thickness by individually optimizing this parameter as well, as is now shown in Figure S16.

In sum, our work presents a new PCE record in Si-filtered CQD PV, and it is supported by a model and confirmed by an extensive suite of experiments that argue against alternative explanations.

Reviewers' comments:

Reviewer #1 (Remarks to the Author):

Thank you for considering the review comments.

The authors have answered almost all the questions and revised the manuscript nicely.

I think no further review is required.

Reviewer #2 (Remarks to the Author):

The revised manuscript has been improved a lot and I appreciate the authors' efforts. However, the following fundamental issues should be made more clearly before it can be recommended to publish in the Nature Communications.

(1) The exciton absorption peak is at 1250 nm bandgap for the pure small bandgap CQD film as shown in Figure S7. But the TA spectra were measured with photoexcitation at 1300 nm, where the photoexcitation photon energy is smaller than the bandgap energy. This is just a special case, since the CQDs are normally photoexcited by photons with an energy larger than their bandgap (exciton absorption energy). Why the authors chose 1300 nm as the photoexcitation wavelength? On the other hand, both small bandgap CQDs and large bandgap CQDs will be photoexcited by using 1160 nm pump light, not only large bandgap CQDs are selectively excited by 1160 nm. Therefore, TA measurements for both small bandgap and large bandgap CQDs should be carried out with both 1160 nm and 1300 nm, respectively. In addition, these TA results under the four conditions should be compared quantitatively with the TA results of the mixed CQD films (Figure S8) to discuss the photoexcited carrier (electron and hole) transfer between the small bandgap QDs and large bandgap QDs.

(2) It is difficult to understand the "charge transfer" between the two different size QDs (i.e. the small bandgap and larger bandgap QDs) from Figure S8. First, "(b) Photoexcitation in the large bandgap population at 1160 nm." In Figure S8 is not true. Small bandgap QDs will also be photoexcited at the same time if the pump light 1160 nm is incident on the mixed QD film other than the large bandgap QDs. Therefore, two TA bleach peaks will be observed at the same time as expected.

(3) The authors think the "charge transfer" occur between the two different size QDs from Figure S8. Then what is the "charge"? Are the "charges" electrons or holes, or both of them? In addition, the authors thought that "charge" can transfer from small bandgap QDs to large bandgap QDs and inversely can transfer from large bandgap QDs to smaller bandgap QDs. However, they also said that "Charge transport is understood to take place through low-bandgap-CQDs percolation channels within the films, as evidenced by the fact that the charge carrier mobility of the mixed CQD films is close to that of the pure low bandgap CQD films (see data in point 2)." These results and discussions are conflict with each other. How do the authors think that the charge transport takes place through low-bandgap-CQDs percolation channels within the films? Is the mobility the electron mobility or hole mobility?

Reviewer #3 (Remarks to the Author):

(Reviewer 3) The revised work of the authors has made improvements regarding on trying to provide supplement experimental data to address the data consistency. However, there is still somewhat lack of information to clearly define the novelty of this work compared to previous reports with similar device structure and materials. Moreover, the impact of the work concerning providing a fundamental

breakthrough in solar cell society is weak.

1)According to previous comment 1, the authors claim that they have cited the previous report, Sci. Rep. 5, 2015, 10626 and appreciated the similar mixing experimental strategy. However, the author didn't clearly define what is their novelty or improvement compared to previous works. For instance, the device performance improvement is more likely due to the better surface passivation other than the advancement embodied in the 'small bandgap difference' mixing.

2)According to previous comment 3, the authors have fabricated 'large bandgap difference' mixture which exhibits expected Voc pinning. However, the authors didn't directly answer the concerns regarding on the possibilities on future improvement. As the question raised previously, the energy difference between two combinations is ONLY 0.08 eV which indicate the space for further combination optimization is extremely small. Is it possible for tweaking the bandgap mixing within 0.08eV?

3)According to previous comment 4, the authors have provided FWHM data regarding on their different combination of QD mixture. However, they didn't perform any rationalized analysis concerning on the 'selection rule' for QD FWHMs. For instance, did the author have any idea what is the best mixture's FWHM to obtain high device performance? What is the requirement for the FWHM regarding on single size QDs? As the authors claimed their main achievement is using a 'small bandgap difference mixing' strategy, the highly monodispersed single size QDs should be the pre-requirements and critical factor. The authors should perform some detailed analysis and comparison with previous works regarding on their size selection rules.

4)According to previous comment 6, the authors have made an effort to identify the energy levels (Fermi and valence band) of a single size and mixture QDs ensembles. However, why in Figure S9 there is no conduction band for the mixture QDs? Especially, where will be the mixture's conduction band? This also introduces another question on how to define the band gap of QDs after mixing? From first exciton peak in absorption spectra?

5)According to previous comment 7, the authors further confirm the lower tail state density is ascribed to the better passivation in small bandgap QDs. Is the surface passivation also having a size-dependent effect?

6)According to previous comment 10, the authors have provided new IQE data stem from a newly equipped instrument. From the data shown in the Figure 4f, the IQE data from three combinations exhibit similar charge collection efficiency. However, if the charge collection efficiency is equal from three combinations, why the device performance (J_{sc}) and EQE show the substantial difference? The authors should provide more discussion to explain these phenomena.

7)According to previous 11, the authors have provided further evidence regarding the 'Au back-mirror effect'. Is this phenomenon can also enhance the device performance under one sun condition? For instance, in Figure S18 the J-V curve of mix QDs devices don't exhibit superior performance compared to single size QD device. However, in Figure 4, the trend is just opposite. Why is this?

Reviewer #4 (Remarks to the Author):

The authors have made a large effort to clarify the critical points of the manuscript as underlined by the reviewers. Overall I find the current version of the manuscript of sufficient quality to be publish, or at least it provide a full overview of the experiments and the interpretation given by the authors.

Still I would like to suggest to include references from other groups than the one of the authors even if their production is massive. I think they can still find the opportunity to cite other groups doing interesting work in the field, between them the group of W. Ma in China, as well as the group of M.A. Loi, G. Konstantatos in Europe, A. Amassian group at Kaust. I think citing others will definitely show that there is a community interested in these topics.

Reviewer #2 (Remarks to the Author):

The revised manuscript has been improved a lot and I appreciate the authors' efforts. However, the following fundamental issues should be made more clearly before it can be recommended to publish in the Nature Communications.

(1) The exciton absorption peak is at 1250 nm bandgap for the pure small bandgap CQD film as shown in Figure S7. But the TA spectra were measured with photoexcitation at 1300 nm, where the photoexcitation photon energy is smaller than the bandgap energy. This is just a special case, since the CQDs are normally photoexcited by photons with an energy larger than their bandgap (exciton absorption energy). Why the authors chose 1300 nm as the photoexcitation wavelength? On the other hand, both small bandgap CQDs and large bandgap CQDs will be photoexcited by using 1160 nm pump light, not only large bandgap CQDs are selectively excited by 1160 nm. Therefore, TA measurements for both small bandgap and large bandgap CQDs should be carried out with both 1160 nm and 1300 nm, respectively. In addition, these TA results under the four conditions should be compared quantitatively with the TA results of the mixed CQD films (Figure S8) to discuss the photoexcited carrier (electron and hole) transfer between the small bandgap QDs and large bandgap QDs.

In the revised text, we now explain more clearly that:

- The choice of 1300 nm was made in the portion of TA studies designed to photoexcite the small-gap phase very selectively. We now explain that Figure S7 shows that, when studying the pure-phase small-gap film as a control, the absorption change at the wavelength corresponding to the large-gap-phase's excitonic feature is at least 20x lower than the absorption change at the wavelength corresponding to the small-gap-phase's excitonic feature.
- The choice of 1160 nm excitation was made in the portion of the TA studies designed to photoexcite the large-gap phase as selectively as possible. We now explain that Figure S7 shows that, when studying the pure-phase large-gap film as control, the absorption change at the wavelength corresponding to the small-gap-phase's excitonic feature is about 5x lower than the absorption change at the wavelength corresponding to the large-gap-phase's excitonic feature.

- We have improved the discussion of Figure S8 to make the conclusions more clear. In particular, referring to S8a, though we directly photoexcited very selectively only the small-gap phase (see Figure S7a), the bleach at 1170 nm – characteristic of the large-gap phase – is within a factor of two of the bleach at 1265 nm. We conclude that charge carriers are able to transfer mildly-uphill from their place of creation (small-gap phase) into the larger-gap phase.

(2) It is difficult to understand the “charge transfer” between the two different size QDs (i.e. the small bandgap and larger bandgap QDs) from Figure S8. First, “(b) Photoexcitation in the large bandgap population at 1160 nm.” In Figure S8 is not true. Small bandgap QDs will also be photoexcited at the same time if the pump light 1160 nm is incident on the mixed QD film other than the large bandgap QDs. Therefore, two TA bleach peaks will be observed at the same time as expected.

We have corrected the caption of Figure S8 to reflect this point. We now more clearly explain that the absorption cross-section is ~13 times greater in the large-bandgap QDs than in the small-bandgap QDs for the case of 1160 nm photoexcitation.

(3) The authors think the “charge transfer” occur between the two different size QDs from Figure S8. Then what is the “charge”? Are the “charges” electrons or holes, or both of them? c In addition, the authors thought that “charge” can transfer from small bandgap QDs to large bandgap QDs and inversely can transfer from large bandgap QDs to smaller bandgap QDs. However, they also said that “Charge transport is understood to take place through low-bandgap-CQDs percolation channels within the films, as evidenced by the fact that the charge carrier mobility of the mixed CQD films is close to that of the pure low bandgap CQD films (see data in point 2).” These results and discussions are conflict with each other. How do the authors think that the charge transport takes place through low-bandgap-CQDs percolation channels within the films? Is the mobility the electron mobility or hole mobility?

The mobility obtained from the FET analysis is the electron mobility, as mentioned in the manuscript. The positive voltage applied to the gate in Figure 2b raises the Fermi level of active layer in the channel that enough electrons create a conductive channel from source to drain.

We agree that transport selectively through the small-gap phase is not unambiguously established in our experiments. We have removed these assertions.

Reviewer #3 (Remarks to the Author):

(Reviewer 3) The revised work of the authors has made improvements regarding on trying to provide supplement experimental data to address the data consistency. However, there is still somewhat lack of information to clearly define the novelty of this work compared to previous reports with similar device structure and materials. Moreover, the impact of the work concerning providing a fundamental breakthrough in solar cell society is weak.

1)According to previous comment 1, the authors claim that they have cited the previous report, Sci. Rep. 5, 2015, 10626 and appreciated the similar mixing experimental strategy. However, the author didn't clearly define what is their novelty or improvement compared to previous works. For instance, the device

performance improvement is more likely due to the better surface passivation other than the advancement embodied in the ‘small bandgap difference’ mixing.

We explain the novelty of the present work over (Sci. Rep. 5, 2015, 10626) as follows:

“We found a regime wherein V_{OC} – rather than being rapidly pinned by the lowest bandgap component in a quantum dot ensemble¹⁸ – is instead related linearly to the bandgap of the ensemble constituents. In this new regime, the V_{OC} for a given bandgap can be increased by the judicious addition of larger bandgap species that modifies the density of states.”

In other words, the previous work (Sci. Rep. 5, 2015, 10626) exploited mixes of two different CQD populations with a bandgap difference of 1 eV, causing the V_{OC} of the ensembles to be pinned by the lowest bandgap component: a V_{OC} of 0.44 V is shown for a mixture of 1.2 eV and 2.2 eV QDs. Therefore, the authors of this work improved the PCE of their devices mainly through J_{SC} . In our work, we select the two CQD components with a small bandgap difference, which gives a near-linear V_{OC} change as function of the ensemble constituents. This strategy improves the J_{SC} at no cost to V_{OC} , leading to an increase in IR PCE of ~11%.

In addition, Figure 4c clearly shows that the mixed CQD devices perform better than both of their single-size constituents, demonstrating that the mixing strategy is responsible for the performance increase, not the better surface passivation.

2)According to previous comment 3, the authors have fabricated ‘large bandgap difference’ mixture which exhibits expected Voc pinning. However, the authors didn’t directly answer the concerns regarding on the possibilities on future improvement. As the question raised previously, the energy difference between two combinations is ONLY 0.08 eV which indicate the space for further combination optimization is extremely small. Is it possible for tweaking the bandgap mixing within 0.08eV?

Further improvement within the 0.08 eV window would indeed be possible, although it would potentially require narrower CQD size distributions and the intermixing of more than two size distributions. One could envision a mix of an arbitrarily high number of single-size CQD distributions which would be tailored to perfectly match the solar spectrum, an extension of the idea we presented in this work. There is therefore potential for future work built upon our results and concept. In addition, the simplicity of the mixing strategy makes it accessible to other CQD PV research groups, especially those working on related tandem applications.

We now explain the potential for future improvement in the discussion:

“This strategy, which allows to decouple the traditional V_{OC} - J_{SC} trade-off, has the potential to raise the IR PCE towards the 6% theoretical limit with the improved light absorption properties of a mix of an arbitrarily high number of single-size CQD distributions perfectly matching the solar spectrum.”

3)According to previous comment 4, the authors have provided FWHM data regarding on their different combination of QD mixture. However, they didn’t perform any rationalized analysis concerning on the ‘selection rule’ for QD FWHMs. For instance, did the author have any idea what is the best mixture’s FWHM to obtain high device performance? What is the requirement for the FWHM regarding on single

size QDs? As the authors claimed their main achievement is using a ‘small bandgap difference mixing’ strategy, the highly monodispersed single size QDs should be the pre-requirements and critical factor. The authors should perform some detailed analysis and comparison with previous works regarding on their size selection rules.

The FWHMs indicate the spectral range that the CQD films can absorb. A greater value of FWHM indicates a larger absorption range, which may result in more J_{SC} if the absorption spectrum is well matched to the solar spectrum. In our work, the largest value of FWHM results in the highest J_{SC} because of the selected CQD size (Figure 2a). To obtain the best performance (PCE), a high V_{OC} is also needed. We chose the small bandgap difference, avoiding V_{OC} loss, thus the PCE is improved. However, the FWHM of single size QDs should indeed be as small as possible to reduce the presence of tail states. Therefore, the largest FWHM provides the best performance in this work, but the conclusion is not suitable in other system, especially for single-size CQDs.

We selected the QD size based on the overlap between the QD absorption spectrum and the solar spectrum. In addition, we choose the QD bandgap difference lower than 0.1 eV to avoid V_{OC} loss. We have explored this parameter space experimentally and the values presented in the manuscript are those which yielded the highest performance; a detailed model and quantitative analysis could indeed be an interesting part of our future work.

We have added the following discussion of the rationale of FWHM selection in the manuscript:

“The EQE of the best mixed CQD device is wider than that of its pure counterparts, as seen by the increase in full-width half-maximum (FWHM) of the exciton peak (Figure S15), which in turn leads to an increase in photocurrent when the absorption spectrum is well matched to the solar spectrum. The shape of the exciton peak and its FWHM was tuned to the solar spectrum as to increase J_{SC} while avoiding V_{OC} loss.”

4)According to previous comment 6, the authors have made an effort to identify the energy levels (Fermi and valence band) of a single size and mixture QDs ensembles. However, why in Figure S9 there is no conduction band for the mixture QDs? Especially, where will be the mixture’s conduction band? This also introduces another question on how to define the band gap of QDs after mixing? From first exciton peak in absorption spectra?

Figure S9 has now been updated with the CB of the mixed QDs, extracted from the position of the first exciton peak in their absorption spectrum.

Figure S9. Energy levels of L, S, and 2-to-1 mixed QD films from ultraviolet photoelectron spectroscopy (UPS). | UPS spectra of L, S, and 2-to-1 mixed QDs (left) and energy levels (Fermi level (E_F) and valence band (VB)) calculated from UPS spectra. A helium discharge source (HeI α , $h\nu = 21.22$ eV) was used and the samples were kept at a take-off angle of 88° . During measurement, the sample was held at a -15 V bias relative to the spectrometer in order to efficiently collect low kinetic-energy electrons. E_F was calculated from the equation: $E_F = 21.22$ eV $-$ SEC, where SEC is the secondary electron cut-off. The difference between valence band (VB) and Fermi level, η , was determined from the VB onset in the VB region. The 1150 and 1250 nm QDs show very similar E_F and VB maxima, matching well with the energy alignment for charge transport between different size QDs. The conduction band (CB) is extracted from the absorption spectra using the first exciton peak.

5) According to previous comment 7, the authors further confirm the lower tail state density is ascribed to the better passivation in small bandgap QDs. Is the surface passivation also having a size-dependent effect?

Yes, the QD surface passivation is size-dependent. As the QD size increases, the polar Pb-rich (111) octahedron surface is reduced to the benefit of the nonpolar (100) cuboctahedron surface, where the surface property changes in term of the transition from the (111) facet to the (100) facet (*JACS*, **2013**, 135, 5278). Both surfaces have different properties and may need different passivation. Additionally, QDs of different sizes may require different ligand surface coverage to minimize agglomeration and preserve size monodispersity.

6) According to previous comment 10, the authors have provided new IQE data stem from a newly equipped instrument. From the data shown in the Figure 4f, the IQE data from three combinations exhibit similar charge collection efficiency. However, if the charge collection efficiency is equal from three combinations, why the device performance (J_{sc}) and EQE show the substantial difference? The authors should provide more discussion to explain these phenomena.

In the revised manuscript, we now better explain that the absorbance, and thus EQE, is higher on average for the mixed system given the similar IQEs for each film, increasing as a result the integrated J_{sc} , calculated from the EQE

$$J_{sc} = \int_0^{\infty} EQE(\lambda) \gamma_i(\lambda) d\lambda ,$$

where $\gamma_i(\lambda)$ is the incident photon flux spectrum – in our case, the AM1.5G solar spectrum. We also now refer the reader to S14 which helps make this clear:

Figure S14. EQE curves and expected J_{sc} integrated under AM1.5G irradiation.

7) According to previous 11, the authors have provided further evidence regarding the ‘Au back-mirror effect’. Is this phenomenon can also enhance the device performance under one sun condition? For instance, in Figure S18 the J-V curve of mix QDs devices don’t exhibit superior performance compared to single size QD device. However, in Figure 4, the trend is just opposite. Why is this?

The ‘Au back-mirror effect’ affects the device absorption spectrum and can not only improve but also decrease the absorption throughout the whole spectral range via constructive or destructive interference of the incident and reflected light. In the full spectral range, the presence of constructive interference peaks is compensated by the presence of destructive interference valleys thus limiting the effect on the photogenerated current. This effect can, however, be harnessed in IR CQD solar cells thanks to their much narrower spectral range; a constructive interference peak is aligned with the exciton peak by tuning the thickness of the active layer, whereas the destructive interference peaks lie outside of the spectral range of interest (~1100 nm – 1400 nm). A quantitative analysis of this phenomenon has been previously published (*ACS Energy Letters*, **2016**, 1 (4), 852-857)

We now expanded the explanation on the effect under one-sun conditions:

“We note that the extended FWHM of the exciton peak did not improve the J_{sc} under one sun condition (Figure 4d and Figure S18), because the light resonance effect affects the device absorption spectrum and can not only improve but also decrease the absorption throughout the whole spectral range via constructive or destructive interference of the incident and reflected light. In the full spectral range, the presence of constructive interference peaks is compensated by the presence of destructive interference valleys, thus limiting the effect on the photogenerated current.”

Reviewer #4 (Remarks to the Author):

The authors have made a large effort to clarify the critical points of the manuscript as underlined by the reviewers. Overall I find the current version of the manuscript of sufficient quality to be publish, or at

least it provide a full overview of the experiments and the interpretation given by the authors. Still I would like to suggest to include references from other groups than the one of the authors even if their production is massive. I think they can still find the opportunity to cite other groups doing interesting work in the field, between them the group of W. Ma in China, as well as the group of M.A. Loi, G. Konstantatos in Europe, A. Amassian group at Kaust. I think citing others will definitely show that there is a community interested in these topics.

We now provide additional references (ref 11, 12, 29) in the revised manuscript.

REVIEWERS' COMMENTS:

Reviewer #2 (Remarks to the Author):

The revised manuscript has been improved a lot and the authors have made a great effort to answer the questions of the reviewers in detail. So I think this manuscript is qualified for publication and I would like to recommend it to be published in the Nature Communications.

Reviewer #3 (Remarks to the Author):

While I was initially very intrigued by the revised manuscript - it doesn't live up to the initial impression after a careful read. I appreciate the authors have put efforts on answering most of the questions. However, as I commented in the past two reports, the overall novelty for Nature Communication of this work seems still unclear. I would happy to recommend this work if the authors could DIRECTLY address following concerns.

1. According to previous comment 1, the device structure used has been published extensively with higher 1-sun efficiencies. The authors appear to cover their lack of efficiency at 1-sun by providing higher PCEs at silicon-filtered NIR irradiance. Unless the authors can explain why this mixed QDs strategy can only be used and important in silicon-filtered NIR irradiance, it can only be assumed that other groups should be able to achieve higher efficiencies at this condition. Specifically, the QDSC improvement is marginal at best (the highest efficiency under one sun is around 9% in this work, while the certified record efficiency of QDSC is 11.28% under the same one sun condition). Overall, this paper doesn't bring any new chemistry, material or device architecture into the community.
2. According to previous comments 2 and 3, in the revised version, the author claim 'one could envision a mix of an arbitrarily high number of single-size CQD distributions which would be tailored to perfectly match the solar spectrum, an extension of the idea we presented in this work'. Did the author perform any theoretical calculations to support this claim? Moreover, if the single-sized QD is paramount important for fabricating solar spectrum matched QDPV, why there is no any relating new chemistry or material works enclosed in the revised version?
3. There are some typo errors in the revised manuscript. For instance, it can be easily found that the title of the manuscript is different with their supporting information.

Reviewer #3 (Remarks to the Author):

While I was initially very intrigued by the revised manuscript - it doesn't live up to the initial impression after a careful read. I appreciate the authors have put efforts on answering most of the questions. However, as I commented in the past two reports, the overall novelty for Nature Communication of this work seems still unclear. I would happy to recommend this work if the authors could DIRECTLY address following concerns.

1. According to previous comment 1, the device structure used has been published extensively with higher 1-sun efficiencies. The authors appear to cover their lack of efficiency at 1-sun by providing higher PCEs at silicon-filtered NIR irradiance. Unless the authors can explain why this mixed QDs strategy can only be used and important in silicon-filtered NIR irradiance, it can only be assumed that other groups should be able to achieve higher efficiencies at this condition. Specifically, the QDSC improvement is marginal at best (the highest efficiency under one sun is around 9% in this work, while the certified record efficiency of QDSC is 11.28% under the same one sun condition). Overall, this paper doesn't bring any new chemistry, material or device architecture into the community.

We have revised the work to make it clear that the purpose of the mixed CQD strategy is to enable improved silicon-filtered operation. Mixing CQDs increases J_{SC} with respect to the large-bandgap CQDs by better matching the absorption spectrum at the exciton peak with the Si-filtered AM1.5 solar spectrum (Figure 3a). We have also added a discussion of how the strategy we use could be further evolved to improve full-spectrum CQD solar cells.

We also note that record 1-sun efficiency has been obtained by using wider-bandgap CQDs (~1.35 eV), whereas we work with IR-absorbing CQDs (< 1eV), which are not optimal for full-spectrum operation.

2. According to previous comments 2 and 3, in the revised version, the author claim 'one could envision a mix of an arbitrarily high number of single-size CQD distributions which would be tailored to perfectly match the solar spectrum, an extension of the idea we presented in this work'. Did the author perform any theoretical calculations to support this claim? Moreover, if the single-sized QD is paramount important for fabricating solar spectrum matched QDPV, why there is no any relating new chemistry or material works enclosed in the revised version?

We do not make this assertion in the manuscript, and indeed it is a subject for future work.

3. There are some typo errors in the revised manuscript. For instance, it can be easily found that the title of the manuscript is different with their supporting information.

We have now corrected the errors in the manuscript and supporting information.